# Impact of numerical choices on water conservation in the E3SM Atmosphere Model Version 1 (EAM V1)

Kai Zhang[1], Philip J. Rasch[1], Mark A. Taylor[2], Hui Wan[1], Ruby Leung[1], Po-Lun Ma[1],
Jean-Christophe Golaz[3], Jon Wolfe[4], Wuyin Lin[5], Balwinder Singh[1], Susannah Burrows[1],
Jin-Ho Yoon[1,*], Hailong Wang[1], Yun Qian[1], Qi Tang[3], Peter Caldwell[3], and Shaocheng Xie[3]

[1]Pacific Northwest National Laboratory, Richland, WA, USA
[2]Sandia National Laboratory, Albuquerque, NM, USA
[3]Lawrence Livermore National Laboratory, Livermore, CA, USA
[4]Los Alamos National Laboratory, Los Alamos, NM, USA
[5]Brookhaven National Laboratory, Upton, NY, USA
[*]Now at: Gwangju Institute of Science and Technology, Gwangju, South Korea

*Correspondence to:* Kai Zhang (kai.zhang@pnnl.gov)

**Abstract.**

The conservation of total water is an important numerical feature for global Earth system models. Even small conservation problems in the water budget can lead to systematic errors in century-long simulations. This study quantifies and reduces various sources of water conservation error in the atmosphere component of the Energy Exascale Earth System Model.

5    Several sources of water conservation error have been identified during the development of the version 1 (V1) model. The largest errors result from the numerical coupling between the resolved dynamics and the parameterized sub-grid physics. A hybrid coupling using different methods for fluid dynamics and tracer transport provides a reduction of water conservation error by a factor of 50 at 1° horizontal resolution as well as consistent improvements at other resolutions. The second largest error source is the use of an overly simplified relationship between the surface moisture flux and latent heat flux at the interface

10    between the host model and the turbulence parameterization. This error can be prevented by applying the same (correct) relationship throughout the entire model. Two additional types of conservation error that result from correcting the surface moisture flux and clipping negative water concentrations can be avoided by using mass-conserving fixers. With all four error sources addressed, the water conservation error in the V1 model becomes negligible and insensitive to the horizontal resolution. The associated changes in the long-term statistics of the main atmospheric features are small.

15    A sensitivity analysis is carried out to show that the magnitudes of the conservation errors in early V1 versions decrease strongly with temporal resolution but increase with horizontal resolution. The increased vertical resolution in V1 results in a very thin model layer at the Earth's surface, which amplifies the conservation error associated with the surface moisture flux correction. We note that for some of the identified error sources, the proposed fixers are remedies rather than solutions to the problems at their roots. Future improvements in time integration would be beneficial for V1.

# 1 Introduction

Mass and energy conservation are two of the basic principles upon which global climate models are constructed, but such conservation can easily be lost when the model equations are discretized for computers. For example, Stevens et al. (2013) reported on several issues in the ECHAM6 atmosphere model that introduced significant changes in the simulated atmosphere
water and energy balance. For the ERA-Interim reanalysis, Berrisford et al. (2011) reported a global moisture residual of 0.003 kg m$^{-2}$ day$^{-1}$ for the period of 1989-2008, equivalent to a spurious sea level drift of 11 cm per century. In our case, a positive water residual was found in an early development version of the Energy Exascale Earth System Model (E3SM) called V1$\alpha$. If converted to precipitation and distributed evenly over the globe, the residual would lead to spurious sea level rise rates greater than 10 cm per century in both the atmosphere-only and coupled model configurations (Table 1). Those errors are substantial
compared with the estimated sea level rise of 17-20 cm in the 20th century (Church and White, 2006; Church et al., 2013). While the relationship between water budget error and sea level drift is not entirely clear, one can argue that the conservation of total water in a coupled climate model system is necessary for a faithful representation of the global and regional water cycle. It is also worth noting that good conservation is achievable if the model parameterizations are formulated carefully (e.g. Zhou et al., 2015) .

The present paper describes our investigation of water conservation error in the E3SM Atmosphere Model (EAM) and documents the modifications in the numerical implementation that led to error reduction by factors of 50 to 100 in model version V1$\beta$ (Table 1). The characteristics of the conservation errors in V1$\alpha$ and their sensitivities to spatial and temporal resolutions are also discussed. We note that the conservation of *energy* is also important for a global climate model and efforts have been made to reduce related errors in EAM and the coupled model system. That work will be presented in a separate
paper.

The remainder of the paper is organized as follows: Section 2 provides an overview of EAM and summarizes the version 0 (V0) and version 1 (V1) configurations. Section 3 describes the four sources of conservation error we have identified so far. The metrics for error diagnosis are defined in Section 4. The magnitude of various error sources and their sensitivities to model resolution are evaluated in Section 5. Section 6 summarizes the findings and points out directions for further work.

# 2 Model overview

E3SM, formerly known as Accelerated Climate Modeling for Energy (ACME), is a global Earth system model developed by the U.S. Department of Energy (DOE) for high-resolution modeling on leadership supercomputing facilities. The model is a descendant of the Community Earth System Model (CESM). The investigation and model improvement discussed in the present paper focus on the atmosphere component EAM.

E3SM V0 is essentially equivalent to CESM1.3_beta10 except for various bugfixes and retuning that have rather small impact on the simulated climate. EAM V0 uses the spectral element dynamical core on a cubed-sphere mesh (Dennis et al., 2012; Taylor and Fournier, 2010) using an explicit Runge-Kutta time integration scheme. The "low resolution configuration" has approximately 1° horizontal resolution, with 30 spectral elements ("ne30") along each edge of the 6 faces of the cube. The

**Table 1.** Water conservation error from the atmosphere model component in the coupled and atmosphere-only (Atm) simulations with E3SM V0, V1$\alpha$ and V1$\beta$. The relative errors are given as the ratio to the global mean precipitation rate, calculated using Eq. (5) in Section 4. The "artificial sea level rise" is defined as an equivalent sea level rise due to the artificial source of water substances in the atmosphere model, calculated using Eq. (6) in Section 4. The results are slightly different if a different length (number of years) is chosen, but they are very similar to the numbers shown in the table.

| Simulation | Simulation Length (year) | Relative Water Conservation Error (%) | Equivalent Sea Level Rise (cm/century) |
|---|---|---|---|
| V0 Atm | 4 | 0.052 | 5.71 |
| V0 Coupled | 28 | 0.051 | 5.48 |
| V1$\alpha$ Atm | 9 | 0.102 | 11.4 |
| V1$\alpha$ Coupled | 99 | 0.139 | 15.8 |
| V1$\beta$ Atm | 9 | 0.00148 | 0.166 |
| V1$\beta$ Coupled | 253 | 0.00171 | 0.188 |
| V1$\gamma$ Atm | 5 | <2.0 e-7 | < 0.002 |

total number of grid cells used for the physics parameterizations is 48602 in a single layer. [1] The number of vertical layers is 30 everywhere, extending from the Earth's surface to about 2 hPa using a pressure-based terrain-following coordinate near the surface and pressure levels near the model top. The key physical processes considered (see Fig. 1a) include deep convection (Zhang and McFarlane, 1995), shallow convection and turbulent transport/vertical diffusion (Park and Bretherton, 2009), cloud microphysics (Morrison and Gettelman, 2008) and macrophysics (Park et al., 2014), aerosol microphysics (Liu et al., 2012), and radiation (Iacono et al., 2008). Most parameterized physical processes use a time step of 30 min; this is also the frequency at which the dynamical core and physics parameterizations are coupled, which we denote by $\Delta t$. Static or dynamic sub-stepping is used by various parameterizations, e.g., cloud microphysics and aerosol activation. Radiation is calculated every model hour.

EAM V1 uses the same dynamical core but the vertical resolution is increased to 72 layers for both the dynamics and physics, with a typical layer thickness of 50–120 m in the bottom 1 km of the atmosphere, except for the lowest model layer which is about 20 m thick. The model top is about 0.1 hPa. New parameterizations in V1 include the Cloud Layers Unified By Binormals parameterization for shallow convection, turbulent transport, and cloud macrophysics (CLUBB, Golaz et al., 2002; Larson et al., 2002; Bogenschutz et al., 2013) and an updated cloud microphysics scheme (MG2, Gettelman and Morrison, 2015). CLUBB and MG2 are sub-cycled together, using 6 sub-steps of 5 minutes each as a default for the 1° horizontal resolution. Other modifications in the sub-grid processes include an updated treatment of aerosol processes (Wang et al., 2013; Liu et al., 2016), the ice cloud microphysics (e.g. Wang et al., 2014), and changes in the default values of uncertain parame-

---

[1] A spectral element contains $3 \times 3 = 9$ quadrilaterals, giving a total of $9 \times 30^2 \times 6 = 48600$ quadrilateral faces at ne30. The parameterizations are calculated at the vertices of the cubed sphere grid. According to Euler's polyhedral formula, the number of vertices on a cubed sphere grid equals the number of quadrilaterals plus 2. Therefore, the total number of grid points for parameterization calculation is 48602.

**Table 2.** Comparison of different atmosphere model configurations discussed in this paper. Abbreviations: Spec. element – spectral-element dynamical of Dennis et al. (2012) and Taylor and Fournier (2010); PB2009 – shallow convection and turbulence parameterization of Park and Bretherton (2009); PBR2014 – cloud macrophysics parameterization of Park et al. (2014); MG1 – stratiform cloud microphysics parameterization of Morrison and Gettelman (2008); MG2 – stratiform cloud microphysics parameterization of Gettelman and Morrison (2015); CLUBB – unified turbulence, shallow convection, and cloud macrophysics parameterization of Golaz et al. (2002) and Larson et al. (2002). L72 – vertical grid with 72 layers. (This is the vertical grid used in the V1 models, while the V0 model uses a 30-layer grid.) Further details of the model configurations can be found in Section 2. The sources of water conservation error are explained in Section 3.

| Model version | V0 | V0_L72 | V0_CLUBB_MG2 | V1$\alpha$ | V1$\beta$ | V1$\gamma$ |
|---|---|---|---|---|---|---|
| **Vertical levels** | 30 layers | 72 layers | 30 layers | 72 layers | 72 layers | 72 layers |
| **Resolved dynamics** | Spec. element | Spec. element | Spec. element | Spec. element | Spec. element | Spec. element |
| **Parameterized physics** | | | | | | |
| Turbulence | PB2009 | PB2009 | CLUBB | CLUBB | CLUBB | CLUBB |
| Cloud macrophysics | PBR2014 | PBR2014 | CLUBB | CLUBB | CLUBB | CLUBB |
| Cloud microphysics | MG1 | MG1 | MG2 | MG2 | MG2 | MG2 |
| **Sources of water conservation error** | | | | | | |
| PDC (Sect. 3.1) | Yes | Yes | Yes | Yes | No | No |
| LHFLX (Sect. 3.2) | N/A | N/A | Yes | Yes | No | No |
| QNEG4 (Sect. 3.3) | Yes | Yes | Yes | Yes | Yes | No |
| QNEG3 (Sect. 3.4) | Yes | Yes | Yes | Yes | Yes | No |
| INTERR (Sect. 3.5) | Yes | Yes | Yes | Yes | Yes | Yes |

ters. A description of EAM V1 can be found at the E3SM website: http://e3sm.hyperarts.com/model/e3sm-model-description/ v1-description/v1-atmosphere/. We note that the so-called "atmosphere-only" simulations of the present climate actually uses an interactive land surface model. From V0 to V1, the land model changed from CLM4.0 (Oleson et al., 2010) to CLM4.5 (Oleson et al., 2013).

5    In the discussions below, we further distinguish three sub-versions of the V1 model: V1$\alpha$ is the configuration this study started with (which was found to have substantial water conservation error, see Table 1). V1$\beta$ and V1$\gamma$ are versions with various improvements in water conservation. Details of the configurations are described in Table 2 and Sections 3, and the results are evaluated in Section 5.

## 3   Sources of water conservation error

10   To help the reader understand the various conservation errors that we discuss later in this section, the sequence of calculation (i.e., the time integration loop) in EAM is depicted in Figure 1a for V0 and Figure 1b for V1. In each flow chart, the blue

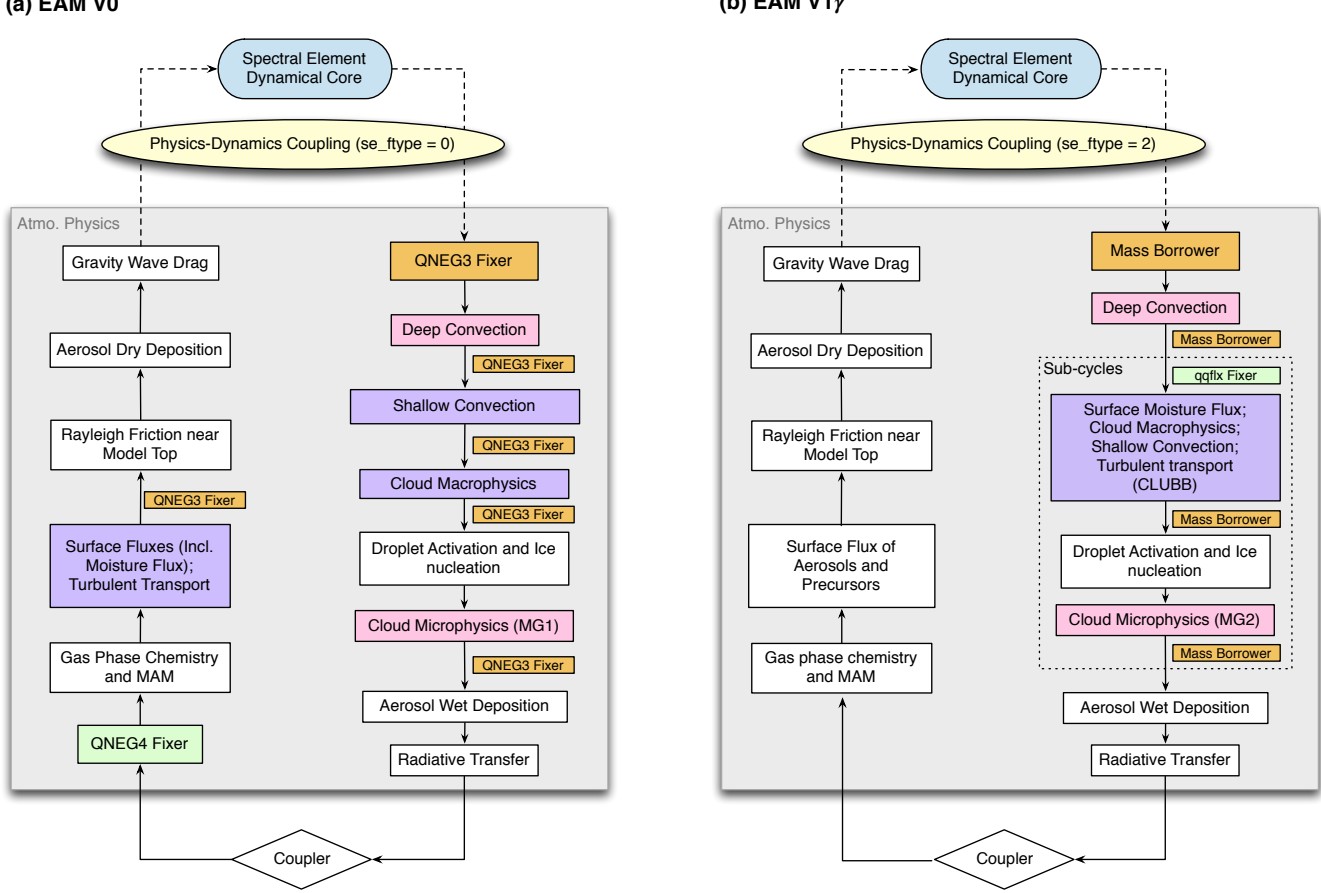

**Figure 1.** Diagrams showing the sequence of calculation (i.e., the time integration loop) in EAM. Left: V0. Right: V1$\gamma$. The blue stadium shapes refer to the resolved-scale dynamics and transport, and the diamonds refer to the exchange of mass and energy with other model components (e.g., land and ocean) through the coupler. The rectangular cells are parts of the physics package that describe the subgrid-scale physical and chemical processes. The colored boxes indicate parts of EAM that affect the concentrations of water species; these include the numerical fixers, deep and shallow convection, turbulent transport, and stratiform cloud macro- and microphysics. The coupling between resolved dynamics and parameterized physics is explained in Section 3.1 and illustrated in Figure 2. The flow chart of V1$\alpha$ is shown in Figure A1.

stadium shape refers to the resolved-scale dynamics and transport, and the diamond refers to the exchange of mass and energy with other model components (e.g., land and ocean) via the coupler. The rectangular cells are parts of the physics package that describe the subgrid-scale physical and chemical processes that operate on a single vertical column. The colored cells indicate parts of EAM that affect the concentrations of water species; these include the numerical fixers, deep and shallow convection,
turbulent transport, and stratiform cloud macro- and microphysics.

Within the physics package, the model uses a sequential splitting method for time integration. This means each parameterized process provides an update of the atmospheric state for the processes the parameterization is responsible for. Each process shown in Figure 1 can use its own time stepping method within a time step $\Delta t$ (e.g., explicit or implicit method, with or without sub-stepping). After a parameterization is calculated, an updated model state is passed on to the next process. The coupling
between the physics package and the resolved dynamics is more complex and is explained in Section 3.1. The coupling between EAM and the other components (e.g., land and ocean) uses sequential split.

Spurious sources and sinks of atmospheric water can result from non-conservative model formulation or discretization choices in the representation of individual physical processes, simplistic algorithms used for removing unphysical (e.g., negative) values, and errors or inconsistencies in the coupling of different physical processes. For EAM, we have identified five
such error sources and four of them are addressed in this study. The details are explained in the subsections below. Each error source is given an acronym to help distinguish the different model configurations summarized in Table 2 and the numerical results presented in Section 5.

## 3.1   Physics-dynamics coupling (PDC)

From a software perspective, the resolved dynamics and parameterized physics form two "packages" of calculations that are
coupled with time interval $\Delta t$. The dynamical core calculates the advection of momentum, heat, air mass, and the mass of additional trace species such as water vapor, cloud condensate, aerosols and their precursors. Within the PDC interval of $\Delta t$, there are two levels of sub-stepping that are relevant to discussions in this paper: At the top level, the entire dynamical core is sub-cycled with se_nsplit (typically 2–4) steps, each containing the calculating of horizontal advection followed by vertical remapping. The horizontal advection is further sub-cycled with se_rsplit (typically 2–3) steps per one vertical remapping. In
the examples shown in Fig. 2, there are se_nsplit = 2 dynamics sub-steps (blue stadium shapes) each containing se_rsplit = 3 horizontal advection steps (green boxes).

Two options are available in V0 and V1$\alpha$ for the coupling between physics and dynamics:

– se_ftype = 1 (Fig. 2a): For all prognostic variables affected by the dynamical core (temperature, winds, surface pressure, and the concentrations of all tracers), the tendencies from the physics package are multiplied by $\Delta t$ to update the
30       atmospheric state before the first dynamics sub-step. When $\Delta t$ is large compared to the characteristic time scales of the parameterized processes, the changes in temperature during a coupling step $\Delta t$ can be sizable but the column-based parameterizations do not provide a mechanism for the wind fields to respond, so spurious gravity waves can be triggered

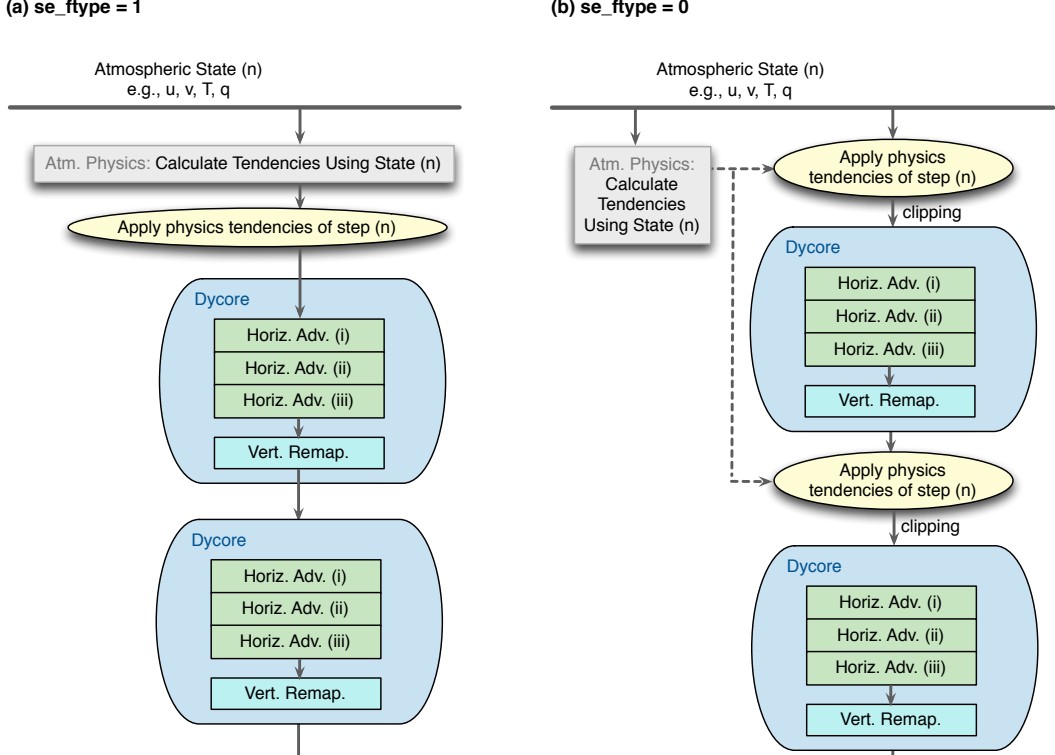

**(a) se_ftype = 1**

**(b) se_ftype = 0**

**Figure 2.** Diagrams showing two options for the coupling between parameterized physics and resolved dynamics in EAM. "Clipping" in panel (b) refers to the fact that negative water concentrations are reset to zero after the physics tendencies are applied. The examples shown here correspond to se_nsplit = 2, se_rsplit = 3. Further details are explained in Section 3.1.

in the dynamics sub-steps. Such gravity waves have been reported in earlier studies, e.g., Lauritzen et al. (2015) and Thatcher and Jablonowski (2016).

– se_ftype = 0 (Fig. 2b): Within a coupling time step $\Delta t$, the physics tendencies are applied as a constant source term in each of the se_nsplit sub-steps. This increases the effective coupling frequency of physics and dynamics, avoiding
5    the spurious gravity waves that could be triggered by se_ftype = 1. For the first dynamics sub-step (the uppermost blue stadium shape in Fig. 2b), the calculation is equivalent to se_ftype = 1 except that a step size of $\Delta t/\text{se\_nsplit}$ (instead of $\Delta t$) is multiplied to the physics tendencies; for the subsequent dynamics sub-steps, the physics tendencies based on step $n$ are inconsistent with the current intermediate atmosphere state, which can lead to situations where the physics tendencies attempt to remove more water than is available in a grid cell. The high-order, locally conservative advection scheme of
10    Dennis et al. (2012) uses a sign-preserving 2D (horizontal) limiter to constrain numerical oscillation (Taylor et al., 2009; Guba et al., 2014). The limiter can prevent negative tracer concentrations, provided that the total concentration in a

spectral element is positive. This condition is fulfilled by se_ftype = 1 and during the first dynamics sub-step of se_ftype = 0. However, for the later dynamics sub-steps of se_ftype = 0, our analysis indicates that the inconsistency between the intermediate model state and the physics tendencies of time step $n$ can lead to negative element-total concentrations that are not anticipated by the transport scheme. In the default V0 and V1$\alpha$ configurations, the negative concentrations are reset to zero after the physics tendencies are applied. As we show in Section 5, this simple "clipping" effectively introduces a spurious source of water mass and is the main source of water non-conservation in these two model versions.

A new physics-dynamics coupling method is used in V1$\beta$ and V1$\gamma$:

- se_ftype = 2: a hybrid method that uses option se_ftype = 0 (Fig. 2b) for the fluid dynamics variables (temperature, winds, and surface pressure), while option se_ftype = 1 (Fig. 2a) is used for water vapor, liquid- and ice-phase condensate, and all other advected tracers. The new option maintains positive semi-definiteness of the tracer concentrations at the physics-dynamics interface (which is the benefit of using se_ftype = 1 for tracers), and meanwhile avoids spurious gravity waves (which is the benefit of using se_ftype = 0 for the dynamics fields). We note that the different treatments between tracer concentrations and the dynamics fields might cause instabilities or other problems, but such issues did not appear in either short (a few days) and long (> 100 years) simulations we have conducted so far.

In addition to this new hybrid method se_ftype = 2, we also have considered other options, such as reducing the overall model physics time step and set se_ftype = 1, which effectively increases the physics-dynamics coupling frequency. This can avoid the spurious gravity waves and maintain the consistent physics-dynamics coupling for tracers and dynamical fields, but the computational cost is substantially higher and we not observe significant improvement of the model simulations. Further investigation is needed to fully understand the pros and cons of using different coupling and time stepping methods.

## 3.2 Inconsistent relationships between surface moisture and latent heat fluxes (LHFLX)

In E3SM, the vertical fluxes of moisture, latent heat, and sensible heat across the Earth's surface are parameterized as part of other components of the Earth System Model (e.g., ocean and land) and provided to EAM by the coupler. The moisture flux is caused by processes like evapotranspiration, fusion, and/or sublimation at the surface, while the latent heat flux is the energy exchange associated with such phase changes. The relationship between the surface moisture flux and latent heat flux can be complex since it depends on the type of the surface and the state of the surface water (liquid or solid). The complexity has been taken into account when deriving the fluxes passed to the atmosphere. However, in the V1$\alpha$ version, an overly simplified relationship is used at the interface between the turbulence/cloud parameterization CLUBB and the host atmosphere model. The moisture flux is derived from the latent heat flux by assuming a constant scaling factor between the two, namely the latent heat of evaporation, and ignoring the distinction from sublimation. The impact of this inconsistency on water conservation is substantial as shown later in Section 5. The resulting conservation error can be removed by directly using the coupler-provided moisture and latent heat fluxes instead of re-derived values.

### 3.3 Clipping of downward moisture fluxes at the Earth's surface (QNEG4)

The surface flux parameterizations can occasionally predict strong downward moisture fluxes. Since EAM uses relatively long time steps for the physics package (30 min at $1°$ resolution), the downward fluxes could lead to negative humidity in the near-surface layers of the atmosphere model, depending on how the surface fluxes are applied in the subsequent calculations, e.g., whether they are applied as an immediate moisture sink in the lowest model layer or as the lower boundary condition in the turbulence scheme, and whether a simple explicit time stepping or a sophisticated implicit and positive semi-definite method is used.

E3SM inherited from its predecessor a subroutine called QNEG4 to guard against negative humidity. The subroutine limits the downward surface moisture flux to an amount that would result in zero water vapor in the lowest layer, assuming (1) the downward flux affects only the bottom layer, (2) no moisture source is provided from layers above, and (3) an Euler forward method is used for time integration. QNEG4 is admittedly a very simplistic and aggressive limiter. Although experience has shown that some amount of adjustment is needed for the downward moisture flux, the actual amount applied by QNEG4 is likely an overestimation since the turbulence parameterizations in both E3SM V0 and V1 use implicit time stepping.

In V0 and its predecessors where the turbulent transport parameterization is calculated immediately after the flux exchanges with the coupler, the QNEG4 limiter is applied after the coupler and before turbulence. When moisture flux is limited, a corresponding adjustment is applied to the latent heat flux assuming the clipped amount of downward moisture flux corresponds to evaporation. The surface sensible heat flux is adjusted by the same amount to ensure that the total (latent plus sensible) flux of energy stays unchanged. However, the moisture fluxes received by the sub-surface components of the Earth system model are unclipped, resulting in an effective moisture source in the coupled system.

In E3SM V1$\gamma$, we replace the QNEG4 limiter by a fixer named QQFLX that borrows water vapor from air aloft to increase humidity in the lowest model layer while keeping the coupler-provided fluxes untouched. The algorithm used by QQFLX is described in detail in Appendix A. Furthermore, since the V1 configurations use CLUBB as the unified parameterization for turbulence, shallow convection, and cloud macrophysics, the surface moisture and heat fluxes (which form lower boundary conditions for the turbulence scheme) are not used to update the atmosphere state until various other parameterized physics process and the resolved dynamics are calculated. This is can be seen in Fig. 1 from the different locations of the light purple boxes in panels (a) and (b). To account for the change in the sequence of calculation, the fixer QQFLX is applied within each of the combined cloud macro- and microphysics sub-step and immediately before CLUBB.

We note that QQFLX determines the amount of water vapor to borrow based on the same 3 assumptions used by QNEG4, hence QQFLX probably also does more work than necessary. Future work on the coupling between surface fluxes and the turbulence parameterization could help to address this issue.

### 3.4 Clipping of negative water concentrations within the physics package (QNEG3)

Apart from the surface moisture flux, the model state tendencies calculated by the other parameterized and resolved processes can also lead to negative water concentration, and the same is generally true for all tracers considered in the model. In V1$\alpha$ and

its predecessors, a subroutine named QNEG3 removes unphysical tracer concentrations during every time step $\Delta t$, both at the beginning of the physics package (i.e., after the resolved dynamics) and after each parameterization. Like the fixers described in Sections 3.1 and 3.3, QNEG3 simply clips the unphysical concentrations to a pre-selected minimum value (typically zero or a small value like $1 \times 10^{-35}$ $\mathrm{kg\,kg^{-1}}$, depending on the tracer). Such clipping also introduces spurious tracer sources.

In V1$\gamma$, a different fixer based on the mass-borrower module of the ECHAM-HAM2 model (Stier et al., 2005; Zhang et al., 2012) is applied, which borrows tracer mass from an adjacent layer and conserves mass. The algorithm used in this alternative fixer is described in Appendix B.

### 3.5   Conservation errors within individual parameterizations (INTERR)

The discretization of individual parameterizations can lead to errors in total water conservation, which we refer to as internal
error in this paper. In the V0 and V1 models, deep convection and cloud microphysics parameterizations are known to produce changes in the column integrated total water that are several orders of magnitude larger than machine rounding. The global averages of these errors are found to be negligible compared to the errors caused by PDC, LHFLX, QNEG3, and QNEG4, thus they are left unaddressed in this study.

### 4   Diagnosing water conservation error

EAM solves time evolution equations of five water species: vapor, stratiform cloud liquid and ice, and large-scale precipitation (rain and snow). For deep convection (and also shallow convection in V0), the convective cloud condensate and precipitation amounts are derived from diagnostic equations; these quantities have no storage, so they are not considered in the water conservation calculation. In the discussions below, we denote the vertically integrated total atmospheric water (unit: $\mathrm{kg\,m^{-2}}$) at time step $n$ in column $i$ by

$W_i^n = W_{i,qv}^n + W_{i,ql}^n + W_{i,qi}^n + W_{i,qr}^n + W_{i,qs}^n$            (1)

where the subscripts $qv$ (vapor), $ql$ (liquid condensate), $qi$ (ice condensate), $qr$ (rain), and $qs$ (snow) refer to the five prognostic water tracers.

According to the conservation law, the change in atmospheric water storage should equal the net water flux through the boundaries plus the total source of water internal to the atmosphere. The column-total water amount at time step $n$ in column
$i$ can be written as

$W_i^n = W_i^{n-1} + (E_i - P_i)\,\Delta t + F_i\,\Delta t + S_i\,\Delta t$            (2)

where $E$ is the surface moisture flux ($\mathrm{kg\,m^{-2}\,s^{-1}}$), $P$ is the precipitation flux ($\mathrm{kg\,m^{-2}\,s^{-1}}$) including rain and snow, $F$ is the column-water tendency caused by resolved advection, and $S$ is the total water tendency within the column caused by local source/sink. In EAM discussed in the present paper, there is no chemical process that produces or consumes water. In the
absence of spurious numerical sources, we have $S_i = 0$, and one can diagnose an "expected" column water amount as follows:

$$W_{i,\text{expected}}^n = W_i^{n-1} + (E_i - P_i)\,\Delta t + F_i\,\Delta t\,. \tag{3}$$

We denote the conservation error of total water in column $i$ and time step $n$ by

$$\Delta W_i^n = W_i^n - W_{i,\text{expected}}^n\,. \tag{4}$$

To get a sense of the physical significance of the conservation error, we calculate the globally averaged total water tendency normalized by the global mean precipitation rate, i.e.,

$$\delta W^n = \frac{\sum_{i=1}^{I} (A_i \Delta W_i^n / \Delta t)}{\sum_{i=1}^{I} (A_i P_i)} = \frac{\sum_{i=1}^{I} (A_i S_i^n)}{\sum_{i=1}^{I} (A_i P_i^n)} \tag{5}$$

where $I$ denotes the total number of grid columns on the cubed sphere mesh. $A_i$ is the grid cell area for column i. $\delta W^n$ is referred to as the normalized relative global mean error in the discussion below. While Eq. (5) gives the conservation error of a

single time step, in Section 5 we will report the temporally averaged values, where the numerator and denominator in Eq. (5) are averaged separately over the selected time window before the division is done.

To demonstrate the long-term impact of the water conservation error, we also report on the global average of Eq. (4) and convert it to an "equivalent sea level change" using

$$\Delta H^n = \frac{1}{\rho_l} \frac{\sum_{i=1}^{I} (A_i \Delta W_i^n)}{\sum_{i=1}^{I} (A_i)}\,. \tag{6}$$

where $\rho_l$ is the density of liquid water. $\Delta H$ is the depth of liquid water that would accumulate at the Earth's surface if the atmospheric water from spurious sources was converted to precipitation and distributed evenly over the surface of the globe. In the actual model, part of the spuriously created water might stay in the atmosphere, resulting in less amount reaching the surface than $\Delta H$, although our analysis suggested that the long-term global-mean net surface flux $(P - E)$ was close to the spurious water source $\Delta W / \Delta t$. Another point to consider is that in reality, the increased precipitation is likely to end up in

the oceans and in reservoirs over land such as lakes, ice caps and glaciers, hence the division in Eq. (6) by the surface area of the entire globe is likely to underestimate the change in sea level. To provide an accurate assessment of the impact of the conservation error, one should conduct a pair of coupled model simulations with and without the fixes discussed in the paper, and compare the simulated sea levels. Unfortunately we did not have sufficient resources to conduct such simulations and evaluate the impact of water conservation in isolation. (The coupled simulations presented in this paper contained various

other changes that have impact on the simulated sea level.) Therefore, the $\Delta H$ reported here should be interpreted as a measure of water conservation error rather than the actual sea level drift in E3SM.

## 5   Simulations and results

This section presents results from the EAM versions listed in Table 2. Although substantial water conservation error was first noticed in V1$\alpha$, we went back to the V0 model and some of its variants to help attribute the errors. That investigation

revealed also the the impact of vertical resolution change on water conservation. The comparisons among V1$\alpha$, V1$\beta$ and V1$\gamma$ demonstrate the impact of the newly implemented fixes. In Section 5.2, we also present results obtained with a subset of the model versions at different spatial (horizontal) and temporal resolutions.

The calculation of conservation error using Eq. (4) is implemented for every individual process in the model, either resolved or parameterized, to allow for a detailed attribution analysis. Our initial explorations revealed that the characteristic error magnitudes can be captured rather accurately using simulations only a few days in length. Therefore 5-day simulations starting from January 1 are presented in this section. Climatological SST and pre-industrial external forcings (e.g. $CO_2$ concentration and solar constant) were used. For a subset of the model configurations, multi-year atmosphere simulations and multi-decade coupled simulations are available. These simulations were also performed under pre-industrial conditions. As we show later in Section 5, results from the long simulations are consistent with those from the short runs.

## 5.1 Magnitudes of different error sources

**Table 3.** Water conservation error in the 5-day atmosphere-only simulations with EAM V0 and V1 model configurations. The equivalent sea level change is calculated using Eq. (6). The Normalized conservation error is calculated using Eq. (5).

| Model version | V0 | V0_L72 | V0_CLUBB_MG2 | V1$\alpha$ | V1$\beta$ | V1$\gamma$ |
|---|---|---|---|---|---|---|
| **Equivalent sea level change per century** | 6.99 cm | 7.90 cm | 13.5 cm | 12.8 cm | 0.127 cm | negligible |
| **Normalized conservation error $\delta W$** | 0.0606% | 0.0776% | 0.120% | 0.128% | 1.26E-3% | negligible |
| Relative contribution of error from different sources | | | | | | |
| PDC | 100% | 99.7% | 77.1% | 74.0% | negligible | (Not calculated) |
| LHFLX | N/A | N/A | 22.8% | 24.7% | N/A | (Not calculated) |
| QNEG4 | 0.00% | 0.282% | 0.00% | 1.24% | 99.8% | (Not calculated) |
| QNEG3+INTERR | 0.00% | 0.029% | 0.0875% | 0.001% | 0.2% | (Not calculated) |

Figure 3 presents the timestep-by-timestep global mean water conservation error in different model versions, shown in the unit of centimeter sea level change per century. The simulations were conducted at $1°$ (ne30) resolution with the default time step size. The corresponding 5-day mean equivalent sea level change, normalized conservation error, and the contribution from different sources are shown in Table 3.

The V0 model contains errors caused by PDC, QNEG4, QNEG3, and the internal errors. In the default configuration, PDC is by far the largest source, contributing to almost 100% of the total error (Figure 3a). There is a non-zero but negligible amount of internal error associated with deep convection (not shown). Although the 5-day mean total error seems rather small compared to the global mean precipitation rate ($\delta W$=0.06%), it would lead to a 7 cm sea level increase when accumulated over a century (Table 3).

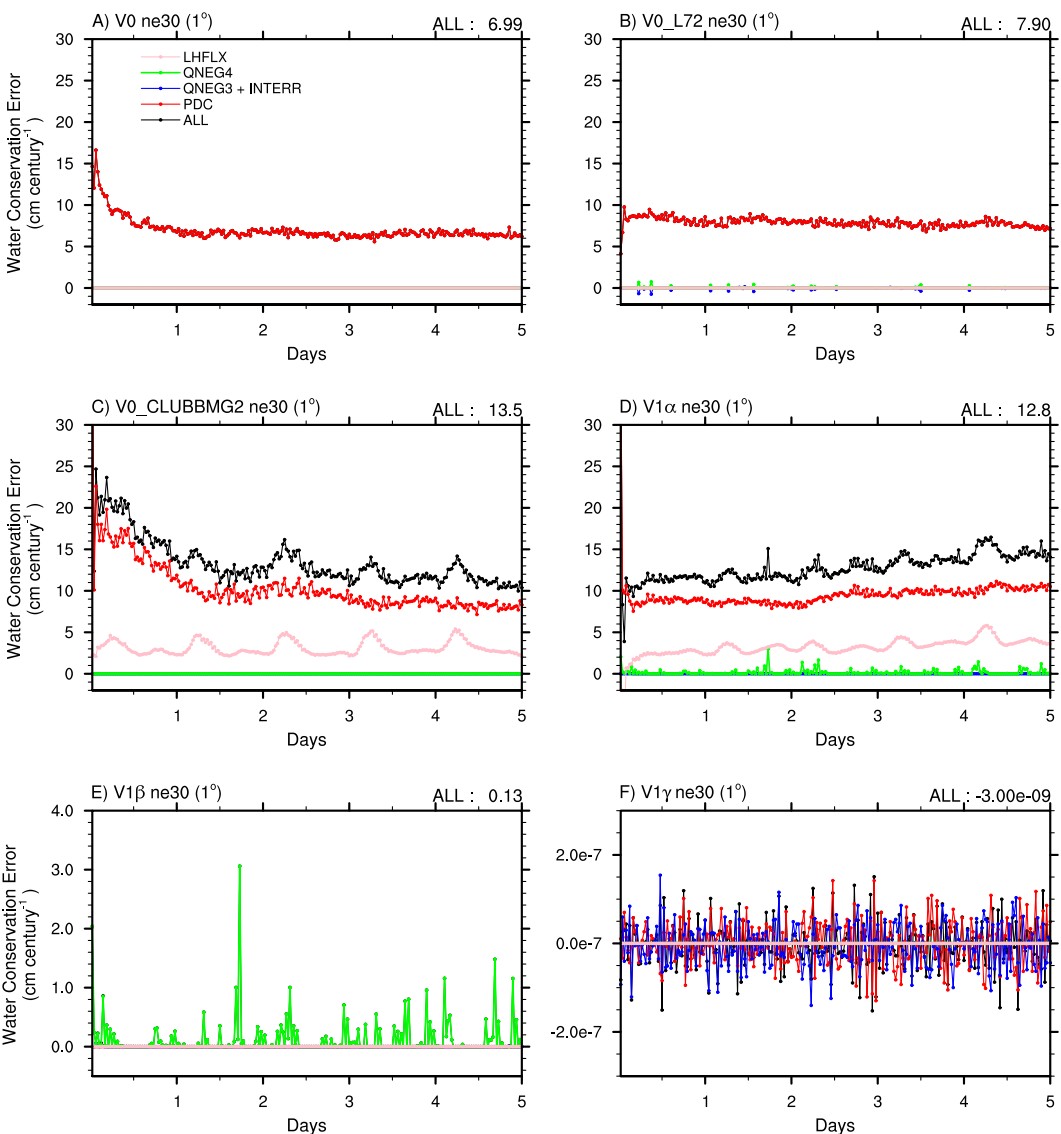

**Figure 3.** Water conservation error in 5-day simulations conducted using different configurations of the E3SM V0 and V1 model at $1°$ (ne30) resolution. The errors are shown as the estimated sea level change (cm) per century, calculated using Eq. (6) at each model time step. The y-axes in the bottom panels (e and f) use scales different from the rest of the figure. The model configurations are summarized in Table 2. The definition of individual sources of water conservation error can be found in Section 3.

Increasing the vertical resolution (V0_L72) leads to an increase in spurious water source associated with the QNEG4 fixer and a slight increase in the PDC error. This is because the 72-level model has a thiner surface layer hence encounters more frequent clipping of the surface moisture flux. The green curve in Figure 3b reveals that the QNEG4 error tends to be sporadic in time. The total errors ($\delta W$ and $\Delta H$) in V0_L72 are about 28% larger than those in V0 (30 vertical layers).

Figure 3c shows the error time series in the V0 model with the original vertical resolution (L30) but with the CLUBB and MG2 parameterizations implemented (V0_CLUBB_MG2). The errors associated with PDC and QNEG4 are similar to those in the default V0 model, and we see another substantial source of error caused by the incorrect relationship between the surface moisture flux and latent heat flux used at the EAM-CLUBB interface (LHFLX). This error amounts to about 23% of the total conservation error. Because of this new error source and the slightly increased PDC error, the total error in the V0_CLUBB_MG2 configuration increases to about twice of that in V0.

The error characteristics in the V1$\alpha$ model shown in Figure 3d are similar to those seen in Figure 3b and 3c, confirming that the non-conservation in this version is caused mainly by PDC, LHFLX, and QNEG4, while the impact from other code changes are small. Compared to V0_L72, the V1$\alpha$ configuration features more frequent occurrence of the QNEG4 error and the magnitudes are somewhat larger. Further analysis shows that positive QNEG4 terms appear mostly over the land areas; therefore the difference is likely attributable to the surface moisture flux changes caused by the update of the land model (which is a major difference in model configuration between V0 and V1$\alpha$). After the two major error sources (PDC and LHFLX) are fixed, the equivalent sea level change is reduced by a factor of 100, from 12.8 cm in V1$\alpha$ to 0.127 cm in V1$\beta$ (Table 3). The further removal of the QNEG4 and QNEG3 errors leads to a version V1$\gamma$ with negligible conservation error. In V1$\gamma$, both the instantaneous errors (Figure 3f) and the 5-day average error are negligible (Table 3).

In terms of geographical distribution (not shown), the PDC errors in V1$\alpha$ systematically occur in cloudy regions with strong horizontal gradient in cloud condensate. The LHFLX errors occur typically in middle and high latitudes due to the lower surface temperature there and the more frequent occurrence of ice sublimation/deposition. The QNEG4 errors mostly occur as isolated and sporadic large values over land, while the QNEG3 and INTERR errors are typically very small and randomly distributed in cloudy regions over the globe.

For the code versions V0, V1$\alpha$, and V1$\beta$, multi-year atmosphere simulations and multi-decade coupled simulations are available. These simulations were not specifically conducted for the water conservation investigation thus did not have detailed diagnostics for the conservation errors from different sources. However, since we have available the instantaneous atmosphere water content at the beginning and the end of each simulation as well as the surface moisture and precipitation flux through the entire duration of the simulations, one can still use Eqs. (3)–(6) to calculate the water conservation error normalized by the mean precipitation rate, as well as the estimated sea level change. The results are shown in Table 1. Consistent with the 5-day simulations discussed above, the water conservation errors in the V1$\alpha$ model are about twice as large as those in the V0 model, while the errors in V1$\beta$ are about two orders of magnitude smaller that those in V1$\alpha$. The other point worth noting is that the results derived from the atmosphere simulations are similar to those from the coupled simulations.

## 5.2 Resolution sensitivity in V1$\alpha$

Having identified the main sources of water conservation error in EAM V0 and V1$\alpha$, the next practical question to answer is how sensitive these errors are to the horizontal resolution and time step size. The target horizontal resolution of the V1 model is 1/4° (ne120). Considering the computational cost of such high resolution, the day-to-day model development typically uses the 1° resolution (ne30) and sometimes even coarser grids (1.9° (ne16) and 2.8° (ne11)). Historically in EAM and its predecessors, the time step sizes used in various parts of the model (e.g. dynamics and physics) might not have been changed proportionally to the change in horizontal resolution; this can complicate the prediction or interpretation of conservation errors in the default configurations at different spatial resolutions.

Three out of the five error sources discussed in Section 3 (i.e., PDC, QNEG4, and QNEG3) are related to simplistic clipping of negative concentrations. Negative concentrations are often caused by a dynamical or physical process predicting a strong sink which, when assumed to persist for a long time, can lead to complete depletion of the tracer. The use of higher temporal resolution (shorter time step) can provide more frequently evaluated sink terms that are more consistent with the current tracer concentration, thus can reduce or eliminate the need for clipping. The use of higher spatial resolution, on the other hand, can lead to sharper spatial gradients in the simulated atmospheric features thus stronger local sinks, potentially leading to more cases that need clipping if the time step size is kept the same.

### 5.2.1 Temporal resolution

To quantify the sensitivity of conservation error to the temporal resolution, simulations were conducted with the V1$\alpha$ configuration at 2.8° (ne11) with a 2 h, 1 h, or 30 min step size for the coupling between physics and dynamics, and with all the sub-step sizes in the model changed proportionally. For example, the number of sub-steps of stratiform cloud macro- and microphysics calculation is kept at 6. Radiation, however, is always calculated every hour. We chose the 2.8° resolution for this group of simulations because of the low computational cost. The simulation length was 5 model days as in most of the 1° (ne30) simulations discussed in Section 5.1.

The time evolution of the conservation errors in the 2.8° simulations is found similar to those in the 1° simulations. After a short spin-up period, the PDC errors become stable and only show time-step-scale variations of very small magnitudes. The LHFLX errors are also very stable, with always positive values and a clear diurnal cycle. These features are consistently seen in simulations conducted with different time step sizes (not shown). The QNEG4 errors appear to be sporadic when time step size is smaller (e.g., 30 min) and occur more often as the time step size is increased (e.g., 1 h and 2 h). The QNEG3 and internal errors remain several orders of magnitude smaller than the others in all these simulations.

The dependence of the error magnitude (5-day mean $\Delta H$) on time step size is shown in Figure 4a for PDC, LHFLX, and QNEG4. The QNEG3 and internal errors are very small thus not shown. The PDC, LHFLX, and QNEG4 errors all show exponential depencies on time step size but with different convergence rates. Here we define the convergence rate as the linear regression coefficient between $\log_{10}(\Delta H)$ and $\log_{10}(\Delta t)$, i.e., the slope of a dashed line in Figure 4. The QNEG4 errors are most sensitive to step size: on average, a factor-of-two change in step size leads to about a factor $\sim$26 change in the

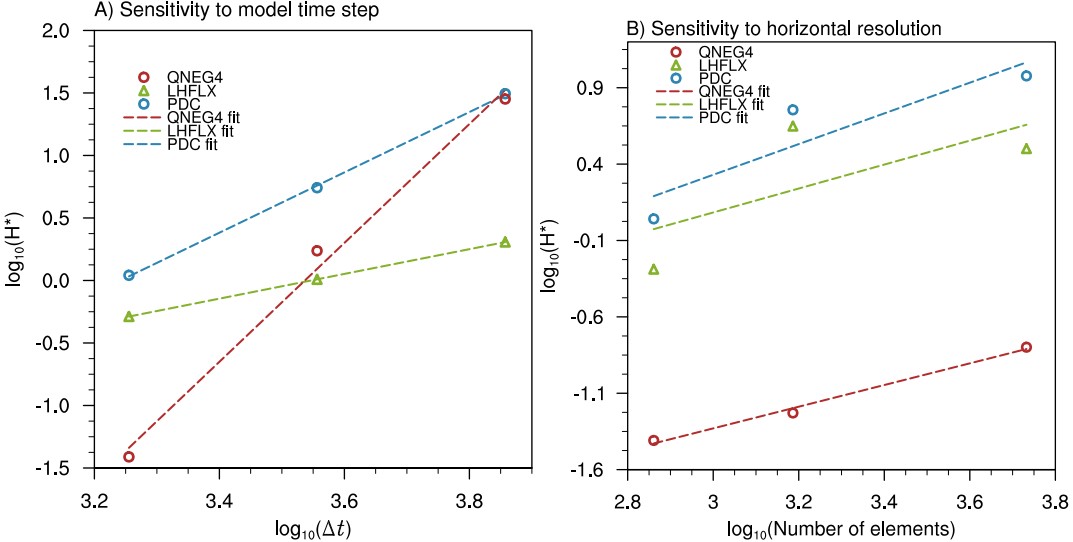

**Figure 4.** Sensitivity of water conservation error to (a) temporal resolution and (b) horizonal resolution in the V1$\alpha$ model. The y-axes in both panels show the global mean sea level change estimated from 5-day simulations using Eq. (6). The unit of $\Delta t$ for the x-axis in panel (a) is seconds. The upper panel shows results conducted at 2.8° (ne11) resolution with different time step sizes. The lower panel shows results using the same time resolution but at 2.8° (ne11), 1.9° (n16), and 1° (ne30) resolutions. Details of the simulation configurations are explained in Sections 5.2.1 and 5.2.2. Only the three largest error sources are plotted in each panel.

conservation error, giving a convergence rate of 4.75. The PDC error converges at a rate of $\sim$2.4, meaning that a factor-of-two step size change corresponds to error change by a factor of $\sim$5. The LHFLX error appears to depend linearly on time step size.

### 5.2.2 Horizontal resolution

When the horizontal resolution increases, we expect the model to be able to resolve finer-scales features and sharper gradients
5  in the horizontal distribution of the water substances, and consequently resulting in stronger local sinks and triggering the fixers more frequently. This very likely will affect the sensitivity of water conservation error to the horizontal resolution. To quantify the sensitivity of conservation error to the horizontal resolution, simulations were conducted with the V1$\alpha$ configuration at 2.8° (ne11), 1.9° (ne16), and 1° (ne30) resolutions using the time stepping configuration of the default 1° model. The physics-dynamics coupling time step was 30 min. There were two vertical remapping time steps for the dynamics, each containing
10  three horizontal advection time steps. The sub-stepping in the parameterized physics were also the same as in the default 1° model.

In this group of simulations, we again see the same time evolution characteristics for the various conservation errors (not shown). The sensitivity of the 5-day mean error magnitude to spatial resolution is shown Figure 4b, plotted as $\log_{10}(\Delta H)$

**Table 4.** Default model configuration parameters for EAM V1$\alpha$ at various spatial resolutions. All configurations use a radiation time step of 1h.

| Parameters | Description | ne11 (2.8°) | ne16 (1.9°) | ne30 (1°) | ne120 (1/4°) |
|---|---|---|---|---|---|
| ne_sphere | Number of spectral elements on the sphere | 726 | 1536 | 5400 | 86400 |
| ncol | Number of physics columns | 6536 | 13826 | 48602 | 777602 |
| dx (km) | Approximate grid box size | 280 | 190 | 100 | 25 |
| se_ftype | Physics-dynamics coupling options | 0 | 0 | 0 | 0 |
| $\Delta t$ | Time step size for physics-dynamics coupling and most physical processes | 7200s | 1800s | 1800s | 900s |
| se_nsplit (time step) | Number of sub-cycles for dynamics (including vertical remapping of the semi-Lagrangian vertical coordinate ) | 4 (1800s) | 1 (1800s) | 2 (900s) | 4 (225s) |
| se_rsplit (time step) | Number of sub-cycles for tracer advection in each dynamics sub-cycle | 2 (900s) | 3 (600s) | 3 (300s) | 3 (75s) |
| cld_macmic_num_steps | Number of sub-cycles for macro-/micro- physics in $\Delta t$ | 6 | 6 | 6 | 6 |

against $\log_{10}(M)$ where $M$ is the total number of spectral elements on the cubed sphere mesh. As expected, the conservation errors generally increase with spatial resolution. A comparison between the two panels of Figure 4 indicates that the sensitivity of the conservation errors to horizontal resolution is weaker than the sensitivity to temporal resolution. The average convergence rate to spatial resolution is 1.0 for PDC, 0.78 for LHFLX, and 0.71 for QNEG4. In addition, a decreased sensitivity to resolution is seen when the horizontal resolution is increased (Figure 4b).

### 5.2.3 Implications

The two groups of resolution sensitivity simulations discussed above indicate that the conservation errors have stronger sensitivities to time step size than to horizontal resolution. We can therefore predict that when the horizontal resolution of the V1$\alpha$

**Table 5.** Water conservation error in 5-day atmosphere-only simulations conducted with the default E3SM V1$\alpha$ model at different horizontal resolutions. The model configurations, including the time step sizes for various parts of the model, are summarized in Table 4. The equivalent sea level change is calculated using Eq. (6). The Normalized conservation error is calculated using Eq. (5).

| Model version | V1$\alpha$_ne11 | V1$\alpha$_ne16 | V1$\alpha$_ne30 | V1$\alpha$_ne120 |
|---|---|---|---|---|
| **Equivalent sea level change per century ($\Delta H$)** | 61.5 cm | 4.56 cm | 12.8 cm | 32.7 cm |
| **Normalized conservation error $\delta W$** | 0.660% | 0.0467% | 0.128% | 0.292% |
| **Contribution to $\Delta H$ from different sources** | | | | |
| PDC | 31.2 cm | 0.00 cm | 9.47 cm | 28.3 cm |
| LHFLX | 2.04 cm | 4.53 cm | 3.16 cm | 4.45 cm |
| QNEG4 | 28.3 cm | 0.0291 cm | 0.159 cm | 0.019 cm |
| QNEG3+INTERR | 0.00 cm | 0.0093 cm | 0.00 cm | 0.01 cm |

model is increased and the time step sizes in various parts of the model are reduced proportionally, the increased temporal resolution will play a dominate role and lead to an overall reduction of the water conservation error.

In reality, however, such proportionality has rarely been applied for the physics parameterizations in our model (Table 4). The time step sizes for various parts of the model are usually chosen empirically based on numerical stability, computational cost, and the evaluation of the model results against observational data. For example, the default V1$\alpha$ model at 1° (ne30) uses a 30 min interval for the coupling of physics and dynamics, a 15 min step size for vertical remapping in the dynamics, and a 5 min step size for the resolved horizontal advection. At 1/4° (ne120) (i.e., with a factor-of-four increase in horizontal resolution), the physics-dynamics coupling interval is reduced only by a factor of two (15 min) but the number of vertical remapping steps for the dynamics is increased by a factor of 2 and the number of horizontal advection steps stays at 3 per remapping step. As a result, the model dynamics sees a time step reduction proportional to the increase in horizontal resolution, but the physics parameterizations are run at relatively longer time steps and the number of dynamics steps per physics step is larger. Not surprisingly, the PDC error becomes a larger contributor to the total error and its absolute magnitude also increases substantially (Table 5). The LHFLX error is similar to that at 1°, while the QNEG4 error decreases due to its very fast convergence with respect to time step size. On the whole, however, the PDC error results in more than doubling of the total conservation error at 1/4° relative to 1°.

At 2.8° (ne11), the default V1$\alpha$ model uses yet another configuration: compared to 1° (ne30), the physics-dynamics coupling time step is increased to 2 hours (a factor-of-4 change), the number of vertical remapping time step is increased, and the number of horizontal advection time step is decreased. The increased physics-dynamics coupling sub-steps and the longer step sizes for the parameterizations lead to considerably larger PDC and QNEG4 errors (Table 5), and a total error about 5 times as large as that at 1°.

From Tables 4 and 5 and the discussions above, it can be seen that in the V1$\alpha$ model, the time step size of the resolved horizontal advection is kept proportional to the horizontal grid spacing while the step size for the physics package is not. Since PDC is the largest source of water conservation error in V1$\alpha$, the varying number of dynamics sub-cycles per physics time step at different horizontal resolutions plays the dominate role in determining the magnitude of total error.

## 5.3 Resolution sensitivity in V1$\beta$ and V1$\gamma$

In Figures 5 and 6, we present the 5-day conservation error time series in V1$\beta$ and V1$\gamma$ at various horizontal resolutions. The time step sizes in the model were configured in the same way as described earlier in Tables 4. Note that both the PDC and the LHFLX errors have been fixed in V1$\beta$ and V1$\gamma$.

QNEG4 is the dominant error source in V1$\beta$. An increase of error with horizontal resolution is seen when comparing Figures 5b and 5c (i.e., 1.9° and 1° both with 30 min time step for the physics package), while the comparison among Figures 5a, 5c, and 5d reveals a clear decrease of error with temporal resolution. Both types of changes are consistent with the results shown earlier in Figure 4.

In V1$\gamma$, the QNEG4 and QNEG3 errors are corrected using mass borrowing algorithms (cf. Sections 3.3 and 3.4), so the remaining errors are expected to come from internal conservation issues within the parameterizations and machine rounding. The time series in Figure 6 indicate that the remaining errors indeed show the characteristics of random noise around zero. There is a slight increase in error as the horizontal grid spacing is reduced, but the overall error magnitudes remain negligible.

## 5.4 Impact of water conservation errors on model climate

As mentioned earlier, the atmosphere-only and coupled E3SM simulations presented in Table 1 were not specifically conducted for the water conservation investigation. Hence, the different model configurations contained not only the water conservation fixes but also code changes in other aspects. In order to isolate the impact of the water conservation errors, we performed three 6-year atmosphere-only simulations: (i) with V1$\alpha$, (ii) with V1$\alpha$ plus the PDC and LHFLX fixes (i.e., similar to V1$\beta$), and (iii) with V1$\alpha$ plus the PDC, LHFLX, QNEG4, and QNEG3 fixes (i.e., similar to V1$\gamma$). The last 5 years of each simulation were used to evaluate the atmospheric features of the simulated climate. The differences between those simulations turned out insignificant compared to the internal variability in the global circulation and the cloud- and precipitation-related statistics, leading to the conclusion that the 5-year climate was unchanged. We did not conduct coupled simulations that differ only in the water conservation fixes, due to the very high computational cost of such simulations. It is expected that effect on the scale of a few years to a couple of decades will be small. For century-long simulations, however, we already showed that the spurious changes in the atmospheric water content can be comparable to the observed level of sea level change. This in turn could affect the water cycle, although details of the impact is not yet known.

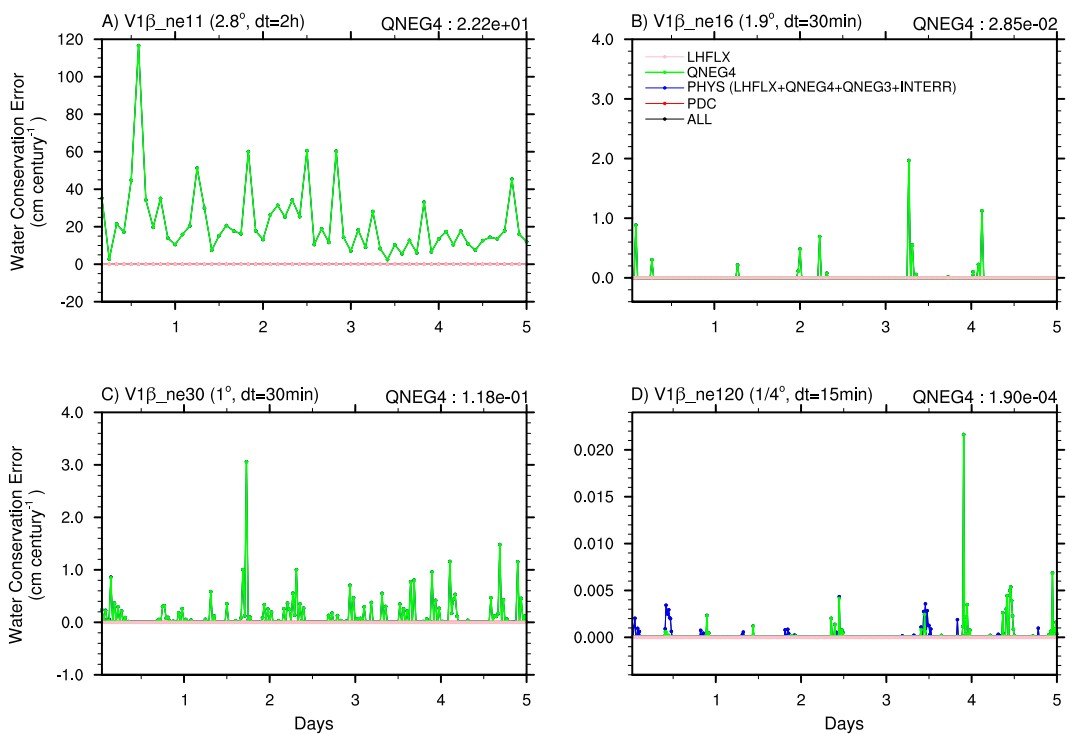

**Figure 5.** Water conservation error in 5-day simulations conducted with the E3SM V1$\beta$ model at different resolutions.

## 6 Conclusions

This study investigated the causes of water conservation problem in E3SM V1$\alpha$ and related model versions. Earlier development of this model and its predecessors already paid substantial attention to conservation in individual resolved or parameterized processes, therefore those errors turned out to be negligible in our simulations. However, less attention had been paid to the impact of the physics-dynamic coupling, formulation inconsistency between different parts of the model, and simplistic artificial fixers that attempt to remove unphysical negative concentrations. As we have shown, those previously overlooked aspects can cause conservation problems as serious as the estimated sea level change in the 20th century.

The four significant error sources we identified in the V1$\alpha$ model fall into two categories: (1) The LHFLX error (Section 3.2) is caused by the use of an overly simplified relationship between the surface moisture flux and latent heat flux at the interface between the host atmosphere model and the new turbulence parameterization. This error can be prevented by applying the same (correct) relationship throughout the entire model. (2) The PDC, QNEG4, and QNEG3 errors (Sections 3.1, 3.3, and 3.4) are caused by the clipping of negative water concentrations which leads to spurious water sources into the model atmosphere. We proposed to avoid the PDC problem by using a different coupling method between the resolved dynamics and parameterized physics, and fix the QNEG4 and QNEG3 problems by using mass borrowers instead of simple clipping.

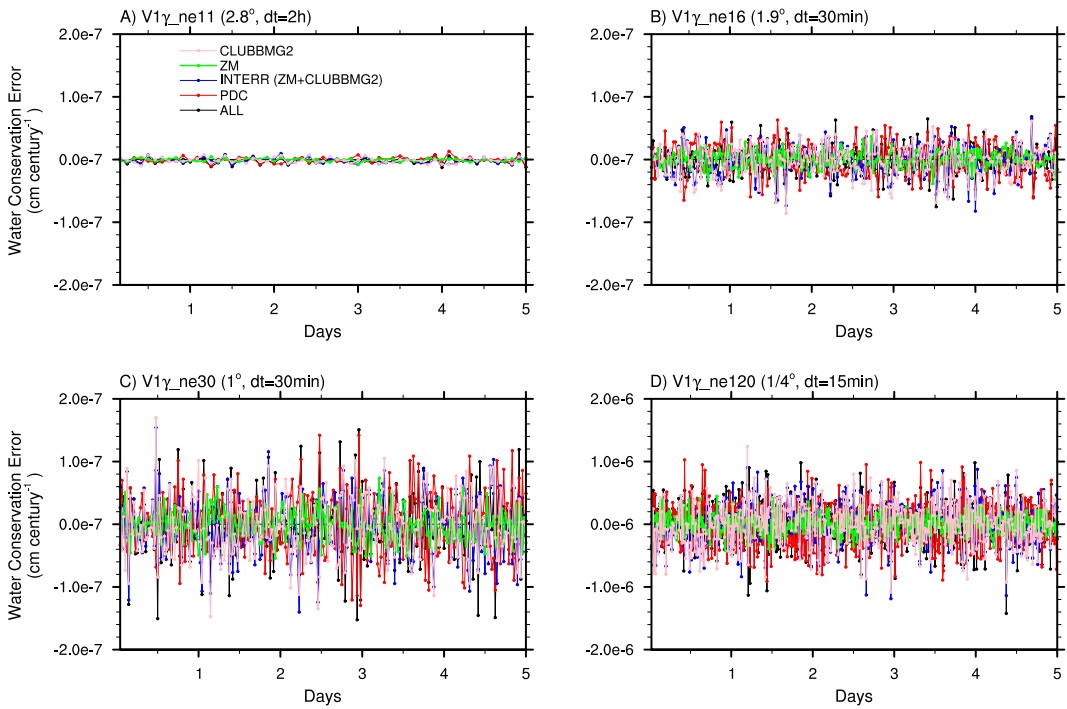

**Figure 6.** Water conservation error in 5-day simulations conducted with the new E3SM V1 configuration (V1$\gamma$) at different resolutions. "ZM" and "CLUBBMG2" indicate the internal water conservation error in the Zhang-McFarlane deep convection scheme and in the CLUBB/MG2 schemes respectively.

Using both short and long, atmosphere-only and coupled Earth system simulations, we showed that PDC is the dominant source of water conservation error in V1$\alpha$, while LHFLX is the largest error source within the physics package. The QNEG4 and QNEG3 errors have considerably smaller magnitudes at 1° or higher resolution. We showed that the elimination of LHFLX and PDC errors reduces the total conservation error by about two orders of magnitude (model version V1$\beta$), and the fixers for

5    QNEG4 and QNEG3 further reduces the errors to a negligible level (version V1$\gamma$).

Sensitivity simulations were conducted to reveal that the magnitudes of the above-mentioned error sources decrease as the model time step size becomes shorter, and increase as the horizontal grid spacing is reduced. The sensitivity to temporal resolution is generally stronger than the sensitivity to horizontal resolution. While one could predict that the water conservation error would decrease if the horizontal and temporal resolutions were increased proportionally, in reality the "standard" configura-

10    tions of the V1$\alpha$ model use empirically chosen time step sizes for the physics package, which leads to varying configurations of physics-dynamics coupling at different resolutions hence non-monotonic changes of water conservation error.

It is worth noting that although we have proposed a model configuration V1$\gamma$ with negligible water conservation error, the strong sensitivity of the results to the physics-dynamics coupling method suggests that the use of long physics time step and dynamics sub-stepping might not be a good idea from the perspective of numerical accuracy. Also, the non-conservation

associated with QNEG3 and QNEG4 in the original model is removed by borrowing mass from other grid cells, which is only a temporary remedy. The fact that negative water concentrations are generated and need to be corrected is another indication that the physics time step is probably too long to provide sufficient time integration accuracy. The very strong time step sensitivity of the PDC, QNEG4, and QNEG3 errors suggests that those errors can be effectively reduced by reducing the physics time

steps. The use of shorter step sizes would have an immediate impact on the computational cost of the model, but the impact on solution integrity needs to be considered (Wan et al., 2013, 2015). Solution accuracy of the time integration strategies, especially in the physics parameterization, deserves further attention.

*Code availability.* The E3SM source code can be obtained from GitHub following the instructions at http://e3sm.hyperarts.com/model/e3sm-code/get-e3sm-code/. The code versions used in this study can be found in the E3SM repository as archival tags: tag "archive/kaizhangpnl/gmd_2016

was used for all the 5-day simulations and the atmosphere-only long simulation for V1$\gamma$; a readme file is available as part of that tag to describe different model configurations used in this study. The long coupled and atmosphere-only simulations shown in table 1 were performed with the following tags:

  – archive/kaizhangpnl/gmd_201705_beta1_v0 for V0;

  – archive/kaizhangpnl/gmd_201605_v1alpha for V1$\alpha$;

– archive/kaizhangpnl/gmd_20161117_beta0 for V1$\beta$.

## Appendix A: The *QQFLX* fixer

If the deposition water vapor over ice surface (downward surface moisture flux) is strong, it might remove all the available moisture in surface layer since the model time step of a global model is often large. This will cause problems in the vertical diffusion calculation, where the surface moisture flux (QFLX) is applied. In the original CESM model, QNEG4 is called to

correct QFLX so that it won't take out all the available moisture from the surface layer. Water conservation is not maintained in the QNEG4 fixer.

The new fixer, named as QQFLX fixer, borrows water vapor from the whole column above the surface layer proportionally and add moisture into the surface layer, so that it can compensate the downward (negative) QFLX.

The excess downward (negative) flux is compared to a theoretical maximum downward flux. The theoretical max is based

upon the given moisture content of lowest level of the model atmosphere.

## Appendix B: The *MassBorrow* fixer

The mass borrower borrows tracer mass from an adjacent layer. It conserves the mass and can avoid negative tracers.

At level k, it will first borrow the mass from the layer k+1 (lower level). If the mass is not sufficient in layer k+1, it will borrow mass from layer k+2. The borrower will proceed this process until the bottom layer. If the tracer mass in the bottom

**EAM V1α**

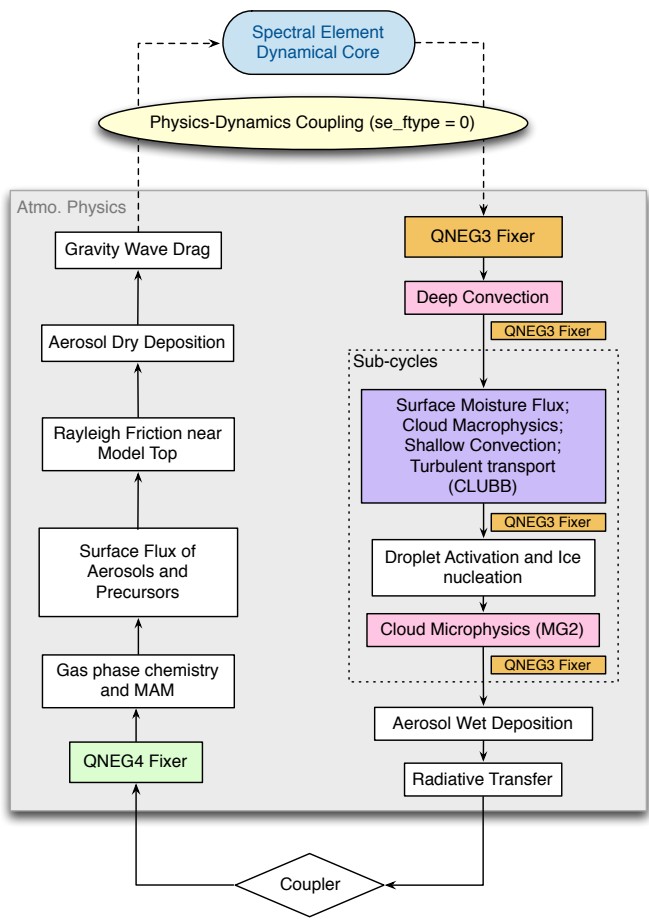

**Figure A1.** Similar to Fig. 1, but for the V1α version of EAM. The diagram for V1β is the same except that the physics-dynamic coupling uses se_ftype = 2.

layer goes negative, it will repeat the process from the bottom to the top. The mass borrower works for any shape of mass profiles, as long as the column integrated mass is positive.

The code is adapted from the tracer mass borrower implemented in the global aerosol-climate model ECHAM-HAM2 (Stier et al., 2005; Zhang et al., 2012).

5 **Appendix C: Time integration loop in EAM V1α.**

Time integration in EAM V1α is depicted in Fig. A1.

*Competing interests.* The authors declare that they have no conflict of interest.

*Acknowledgements.* This research was supported as part of the Energy Exascale Earth System Model (E3SM) project, funded by the U.S. Department of Energy (DOE), Office of Science, Office of Biological and Environmental Research. The authors thank all E3SM team members for their efforts in developing and supporting the E3SM model. We thank Andrew Gettelman for providing the updated code
5 of the water/energy conservation check for MG2 microphysics and Erika Roesler for performing test simulations. We also thank Anthony Craig and Peter Laurizen for helpful discussions. Pacific Northwest National Laboratory (PNNL) is operated for DOE by Battelle Memorial Institute under contract DE-AC06-76RLO 1830. Work at LLNL was performed under the auspices of the US DOE by Lawrence Livermore National Laboratory under contract No. DE-AC52-07NA27344. J-H Yoon was partially supported by the National Research Foundation grant NRF-2017R1A2B4007480. This research used high-performance computing resources from the Oak Ridge Leadership Computing Facility
10 (OLCF) at the Oak Ridge National Laboratory, supported by the Office of Science of DOE under contract no. DE-AC05-00OR22725, the PNNL Institutional Computing (PIC), and the National Energy Research Scientific Computing Center (NERSC), a DOE Office of Science User Facility supported by the Office of Science of the U.S. Department of Energy under Contract No. DE-AC02-05CH11231.

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
