# Peer review of "Impact of numerical choices on water conservation in the E3SM Atmosphere Model Version 1 (EAM V1)"

_Geoscientific Model Development, 2017_

## Short Comment (SC1) · 22 Dec 2017

"Code Availability" does not outline any mean for the reader to obtain access to the program code as outlined in https://www.geoscientific-model-development.net/about/manuscript_types.html. The statement "The E3SM model source code is expected be released to the public in early 2018" is not sufficient. The authors may consider to hold the publication of the manuscript until the relevant version of E3SM has been released.

Lutz Gross GMD Executive Editor.

[Figure]

2017.

---

## Author Comment (AC1) · 24 Jan 2018

We thank the Executive Editor Lutz Gross for the comment and suggestion. After consulting with the E3SM executive committee and discussing with our Topical Editor Dr. Paul Ullrich, we would like to follow the recommendation to hold the publication of the final manuscript until E3SM version 1 is released.

---

## Referee Comment (RC1) · Anonymous Referee #1 · 14 Feb 2018

The authors of this paper found the sources of water conservation error in E3SM atmosphere model that leads to long-term sea level rising and proposed the remedies to resolve them. This paper describes the error sources and fixing methods, as well as provides the sensitivity analysis of the water conservation error to model resolutions. Conservation is one of the most important issues scientists should pay attention to when developing the model. It is a hidden threat to long-term simulation. Although the fixing methods in this manuscript are somewhat remedies and bugfixes instead of root cures, the contribution this manuscript represents is an important achievement to E3SM development. The solution this manuscript proposed can be applied to any other model. The manuscript is well-organized and the conclusion is convincing. I

would suggest accepting with revisions based on the following comments.

Specific Comments

In the introduction section, there is little to no evidence/literature showing the relationship between water conservation error and sea level rising. The literature mentioned is too weak to support this connection. The authors may need to provide some strong evidence on it.

Equation (6): The meaning of "W" is vertically integrated total atmospheric water with a unit of kg/mˆ2. After multiplying "A", the grid cell area, and dividing by liquid water density, the result should be volume, but not height. I think the area of ocean is missing in the equation. Assume this equation is corrected. It is possible that some local spurious water source/sink stay in the atmosphere, leading to less sea level rising. So, it may not have that large effect on the actual sea level rising.

Technical Corrections:

Line 18: I am not sure whether you can cite an unpublished paper: Rasch et al, 2017.

---

## Referee Comment (RC2) · Anonymous Referee #2 · 21 Feb 2018

Review of: Impact of numerical choices on water conservation in
the E3SM Atmosphere Model Version 1 (EAM V1)
by: Kai Zhang, et al.
gmd-2017-293

I recommend publication after addressing concerns cited below.
The paper is a useful description of water conservation errors
in EAM V1.

Suggested word changes; if these suggestions are not correct
then there are more serious problems that I do not understand in
this paper.

1-16: ".. errors in early V1 versions decrease .."
1-17: "Increased vertical resolution in V1 results .."
1-20: ".. beneficial for V1."

2-18: ".. errors in V1$\alpha$ and .."

3-3: " .. 30 everywhere .."  I assume this model does not use
step-mountain.
3-15: ".. 6 sub-steps of 5 minutes each as .."

5-1: "..chemical processes that operate on a single vertical
column."
5-7: "..passed on .."

6-11: ".. which includes advection of air mass, momentum, and
heat."

8-20: ".. evaporation and ignoring the heat of sublimation."
8-32: ".. clipped amount of downward moisture .."

9-2: ".. received by the sub-surface components .."
9-3: ".. are unclipped, .."
9-9: "This can be seen .."
9-20: ".. used in this alternative .."

17-5: ".. advection time steps of 5 minutes for each vertical
remapping step."
17-6: ".. increase in linear horizontal resolution)," or "..
increase in $\Delta x$),
17-8: ".. advection steps stays at 3 per remapping step."
17-14: ".. increased to 2 hours, 4 times that of 1$^\circ$ .."

Confusing things.
3-2: 90*90*6 = 48600.  Why are there 2 additional columns on the cube sphere grid.

3-12: How thick are the 14 layers over mountains?

6-Fig2: It should be mentioned that this diagram shows processes of the stadium shaped cells of Figure 1, that (a) relates to se_type = 1, and (b) relates to se_type 0.  From what I understand, that are 3 se_rsplit steps for each se_nsplit which makes the diagram misleading.  I suggest you discard Figure 2 and explain its contents in the text.

6-11 to 7-3: Clean this up.  According to Table 4 there are 2 to 3 r steps for each n step.  Make this clear.  Mention that humidity advection is grouped with dynamical advection or with tracers depending on model version.

7-22 to 7-24: Does this sentence apply to all se-ftypes?  If so, move it.

8-27: Say more about the time stepping method.  Is it explicit ?  If it is implicit, is it intelligent enough no recognize the availability of water vapor in the boundary layers and not just the lowest layer?

9-22: Do these errors occur consistently over mountain tops of at ocean cells adjacent to mountains, or are they more sporadic? Same question applies to other sub-sections of Section 3.

10-27: It appears that your water mass change is distributed uniformly over the globe.  This water mass mainly affects the ocean fraction of the Earth.  Should your sea-level changes be computed as $\Delta H / (1 - OceanFraction)$.  Please comment on this.

12-Figure3: Most noticeable in the lower left panel, do these spikes originate from large errors at single grid cells? Although water mass borrowing has eliminated the spikes in model V1$\gamma$, the original errors are still there.

17-14 to 17-16: "increased" or "decreased" relative to what. Perhaps this sentence should start: "Comparing V1$\alpha$ model at 2.8° versus that at 1°: the .."

---

## Author Comment (AC3) · 27 Mar 2018

**Reply to Reviewer #2**

March 27, 2018

We greatly appreciate the referee's careful and insightful review. Our responses are detailed below.

*I recommend publication after addressing concerns cited below. The paper is a useful description of water conservation errors in EAM V1.*

*Suggested word changes. if these suggestions are not correct then there are more serious problems that I do not understand in this paper.*

- *1-16: ".. errors in early V1 versions decrease .."*

- *1-17: "Increased vertical resolution in V1 results .."*

- *1-20: ".. beneficial for V1."*

- *2-18: ".. errors in V1$\alpha$ and .."*

- *3-3: " .. 30 everywhere .." I assume this model does not use step-mountain.*

- *3-15: ".. 6 sub-steps of 5 minutes each as .."*

- *5-1: "..chemical processes that operate on a single vertical column."*

- *5-7: "..passed on .."*

- *6-11: ".. which includes advection of air mass, momentum, and heat."*

- *8-20: ".. evaporation and ignoring the heat of sublimation."*

- *8-32: ".. clipped amount of downward moisture .."*

- *9-2: ".. received by the sub-surface components .."*

- *9-3: ".. are unclipped, .."*

- *9-9: "This can be seen .."*

- *9-20: ".. used in this alternative .."*

- *17-5: ".. advection time steps of 5 minutes for each vertical remapping step."*

- *17-6: ".. increase in linear horizontal resolution)," or ".. increase in Dx),*

- *17-8: ".. advection steps stays at 3 per remapping step."*

- *17-14: ".. increased to 2 hours, 4 times that of 1 .."*

The suggested wording changes are applied in the revised manuscript.

*Confusing things.*

*3-2: 90\*90\*6 = 48600. Why are there 2 additional columns on the cube sphere grid.*

The physical parameterizations are computed at the vertices (instead of quadrilateral faces) of the cubed sphere grid. 48600 ($90\times90\times6$) is the number of quadrilaterals, while $48600 + 2 = 48602$ is the number of vertices. The relationship between these two numbers can be derived using Euler's polyhedron formula. For example, the cube has 6 faces, and $6 + 2 = 8$ vertices. This is explained in the revised manuscript.

*3-12: How thick are the 14 layers over mountains?*

The original sentence in the discussion paper reads "...the vertical resolution is increased to 72 layers for both the dynamics and physics, with about 14 levels between the surface and 850 hPa." We replaced the second half of the sentence by "with a typical layer thickness of 50–120 m in the bottom 1 km of the atmosphere, except for the lowest model layer that is about 20 m thick". These new numbers are valid both over the ocean and over the mountains.

*6-Fig2: It should be mentioned that this diagram shows processes of the stadium shaped cells of Figure 1, that (a) relates to se_type = 1, and (b) relates to se_type 0. From what I understand, that are 3 se_rsplit steps for each se_nsplit which makes the diagram misleading. I suggest you discard Figure 2 and explain its contents in the text.*

Both Fig. 1 and Fig. 2, as well as the text, are revised to address this comment.

*6-11 to 7-3: Clean this up. According to Table 4 there are 2 to 3 r steps for each n step. Make this clear. Mention that humidity advection is grouped with dynamical advection or with tracers depending on model version.*

The corresponding paragraph is rewritten to describe the two levels of sub-stepping. We point out during the description of se_ftype =2 that water vapor is grouped with other tracers (e.g., cloud

condensate and aerosols).

*7-22 to 7-24: Does this sentence apply to all se-ftypes? If so, move it.*

Yes, the sentence describes a feature of the transport scheme and is valid for all se_ftypes. We clarify in the revised manuscript that the condition of positive element-total concentration is fulfilled by se_ftype = 1 and for the first dynamics sub-step of se_ftype = 0, but can be violated for the later dynamics sub-steps when se_ftype = 0 is used, resulting in a situation that is not anticipated by the transport scheme.

*8-27: Say more about the time stepping method. Is it explicit? If it is implicit, is it intelligent enough no recognize the availability of water vapor in the boundary layers and not just the lowest layer?*

We agree with the referee that how the downward moisture flux affects near-surface humidity depends on how the flux is applied in the subsequent calculations. In the revised manuscript, we point out that QNEG4 assumes (1) the downward flux affects only the bottom layer, (2) no moisture source is provided from the layers aloft, and (3) an Euler forward method is used for time integration. QNEG4 is admittedly a very simplistic and aggressive limiter. Although experience has shown that *some* amount of adjustment in the downward moisture flux is needed, the actual amount applied by QNEG4 is likely an overestimation since the turbulence parameterizations in both E3SM V0 and V1 uses implicit time stepping methods. We also added text in the same subsection to note that the new QQFLX fixer determines the amount of water vapor to borrow based on the same 3 assumptions used by QNEG4, hence QQFLX probably also does more work than necessary. Future work on the coupling between surface fluxes and the turbulence parameterization could help to address this issue.

*9-22: Do these errors occur consistently over mountain tops of at ocean cells adjacent to mountains, or are they more sporadic? Same question applies to other sub-sections of Section 3.*

In terms of geographical distribution, the PDC errors in V1$\alpha$ systematically occur in cloudy regions with strong horizontal gradient in cloud condensate. The LHFLX errors occur typically in middle and high latitudes due to the lower surface temperature there and the more frequent occurrence of ice sublimation/deposition. The QNEG4 errors mostly occur as isolated and sporadic large values over land, while the QNEG3 and INTERR errors are typically very small and randomly distributed in cloudy regions over the globe. A paragraph is added to Section 5 ("Simulations and results") to describe the above-mentioned features. We did not see frequent occurrences of those errors over mountain tops or at ocean cells adjacent to mountains.

*10-27: It appears that your water mass change is distributed uniformly over the globe. This water mass mainly affects the ocean fraction of the Earth. Should your sea-level changes be computed as $\Delta H$ / (1 - OceanFraction). Please comment on this.*

The water mass change we report in the paper is indeed uniformly distributed over the globe. In reality, the mass change is likely to end up in the oceans and in reservoirs over land such as lakes, ice caps and glaciers, hence our formula (Eq. 6) is likely to underestimate the resulting change in sea level. To provide an accurate assessment of the impact of the conservation error, one should conduct a pair of coupled model simulations with and without the fixes discussed in our paper, and compare the simulated sea levels. Unfortunately we did not have sufficient resources to conduct such simulations to evaluate the impact of water conservation in isolation. (The coupled simulations presented in the paper were not specifically conducted for the water conservation investigation, and therefore contained various other code changes that had impact on the simulated sea level.) The "equivalent sea level rise" reported in our paper is essentially a measure of water conservation error, not the actual sea level drift in E3SM. This is clarified in the revised manuscript after Eq. (6) is presented.

*12-Figure3: Most noticeable in the lower left panel, do these spikes originate from large errors at single grid cells? Although water mass borrowing has eliminated the spikes in model V1γ, the original errors are still there.*

Yes, the spikes seen in the QNEG4 error originate from large errors at single grid cells over land. This is mentioned in a new paragraph added to Section 5 (see also our response to the referee's comment on line 9-22 above).

The last sentence of the abstract and the last paragraph in the conclusions section of our paper both point out that the proposed fixers are remedies rather than root cure of the conservation problems. Future improvements in the time integration methods would be beneficial for the V1 model.

*17-14 to 17-16: "increased" or "decreased" relative to what. Perhaps this sentence should start: "Comparing V1a model at 2.8° versus that at 1°: the .."*

The comparison was made with respect to the 1° simulation. The corresponding sentence is revised in the new manuscript.

---

## Author Response (AR1)

Dear Editor,

Thank you for handling our manuscript. We hereby submit a revised version of the paper. Below please find the point-by-point responses to the reviewer comments, the corresponding changes in the manuscript, and a marked-up version of the revised paper.

We look forward to your favorable decision regarding the revision. Sincerely,

Kai Zhang

on behalf of all co-authors

**Reply to Reviewer #1**

April 12, 2018

The referee's insightful comments are greatly appreciated. Our responses are detailed below.

*The authors of this paper found the sources of water conservation error in E3SM atmosphere model that leads to long-term sea level rising and proposed the remedies to resolve them. This paper describes the error sources and fixing methods, as well as provides the sensitivity analysis of the water conservation error to model resolutions. Conservation is one of the most important issues scientists should pay attention to when developing the model. It is a hidden threat to long-term simulation. Although the fixing methods in this manuscript are somewhat remedies and bugfixes instead of root cures, the contribution this manuscript represents is an important achievement to E3SM development. The solution this manuscript proposed can be applied to any other model. The manuscript is well-organized and the conclusion is convincing. I would suggest accepting with revisions based on the following comments.*

*Specific Comments*

*In the introduction section, there is little to no evidence/literature showing the relationship between water conservation error and sea level rising. The literature mentioned is too weak to support this connection. The authors may need to provide some strong evidence on it.*

The relationship between water conservation error and sea level rising is indeed not entirely clear. The revised introduction acknowledges this, and puts the emphasis of discussion on conservation error instead of sea level prediction. We clarify that the reported "equivalent sea level rise" is the depth of liquid water that would accumulate at the Earth's surface if the increased water amount in the atmosphere were converted to precipitation and distributed evenly over the surface of the globe. Below is the revised wording in the introduction section:

*"For the ERA-Interim reanalysis, Berrisford et al. (2011) reported a global moisture residual of 0.003 $kg\,m^{-2}\,day^{-1}$ for the period of 1989-2008, equivalent to a spurious sea level drift of 11 cm per century. In our case, a positive water residual was found in an early development version of the Energy Exascale Earth System Model (E3SM) called V1$\alpha$. If converted to precipitation and distributed evenly over the globe, the residual would lead to spurious sea level rise rates greater than 10 cm per century in both the atmosphere-only and coupled model configurations (Table 1).*

*Those errors are substantial compared with the estimated sea level rise of 17-20 cm in the 20th century (Church and White, 2006; Church et al., 2013). While the relationship between water budget error and sea level drift is not entirely clear, one can argue that the conservation of total water in a coupled climate model system is necessary for a faithful representation of the global and regional water cycle."*

*Equation (6): The meaning of "W" is vertically integrated total atmospheric water with a unit of kg/m^2. After multiplying "A", the grid cell area, and dividing by liquid water density, the result should be volume, but not height. I think the area of ocean is missing in the equation. Assume this equation is corrected. It is possible that some local spurious water source/sink stay in the atmosphere, leading to less sea level rising. So, it may not have that large effect on the actual sea level rising.*

Thanks for pointing this out. Equation (6) in the discussion paper is missing a denominator, and what we used was the surface area of the entire globe. The reported "equivalent sea level rise" is the depth of liquid water that would accumulate at the Earth's surface if the atmospheric water from spurious sources was converted to precipitation and distributed evenly over the surface of the globe. As the referee pointed out, part of the spuriously created water might stay in the atmosphere, resulting in less amount reaching the surface than we reported, although our analysis suggested that the long-term global-mean net surface flux $(P - E)$ was close to the spurious water source $\Delta W / \Delta t$. Another point to consider is that in reality, the increased precipitation is likely to end up in the oceans and in reservoirs over land such as lakes, ice caps and glaciers, hence the division in Eq. (6) by the surface area of the entire globe is likely to underestimate the change in sea level. To provide an accurate assessment of the impact of the conservation error, one should conduct a pair of coupled model simulations with and without the fixes discussed in our paper, and compare the simulated sea levels. Unfortunately we did not have sufficient resources to conduct such simulations to evaluate the impact of water conservation in isolation. (The coupled simulations presented in the paper contained various other changes that have impact on the simulated sea level.) The "equivalent sea level rise" reported in our paper is essentially a measure of water conservation error, not the actual sea level drift in E3SM. This is clarified in the revised manuscript after Eq. (6) is presented:

*"To demonstrate the long-term impact of the water conservation error, we also report on the global average of Eq. (4) and convert it to an "equivalent sea level change" using*

$$\Delta H^n = \frac{1}{\rho_l} \frac{\sum_{i=1}^{I} (A_i \Delta W_i^n)}{\sum_{i=1}^{I} (A_i)}. \tag{6}$$

*where $\rho_l$ is the density of liquid water. $\Delta H$ is the depth of liquid water that would accumulate at the Earth's surface if the atmospheric water from spurious sources was converted to precipitation and distributed evenly over the surface of the globe. In the actual model, part of the spuriously created water might stay in the atmosphere, resulting in less amount reaching the surface than $\Delta H$, although our analysis suggested that the long-term global-mean net surface flux $(P - E)$*

*was close to the spurious water source $\Delta W / \Delta t$. Another point to consider is that in reality, the increased precipitation is likely to end up in the oceans and in reservoirs over land such as lakes, ice caps and glaciers, hence the division in Eq. (6) by the surface area of the entire globe is likely to underestimate the change in sea level. To provide an accurate assessment of the impact of the conservation error, one should conduct a pair of coupled model simulations with and without the fixes discussed in the paper, and compare the simulated sea levels. Unfortunately we did not have sufficient resources to conduct such simulations and evaluate the impact of water conservation in isolation. (The coupled simulations presented in this paper contained various other changes that have impact on the simulated sea level.) Therefore, the $\Delta H$ reported here should be interpreted as a measure of water conservation error rather than the actual sea level drift in E3SM."*

*Line 18: I am not sure whether you can cite an unpublished paper: Rasch et al, 2017.*

We replaced this citation by a reference to the "V1 Description" section of the E3SM public website:

*"A description of EAM V1 can be found at the E3SM website: http://e3sm.hyperarts.com/model/e3sm-model-description/v1-description/v1-atmosphere/"*

The URL provided above is currently password-protected but will become public with the release of E3SM (expected: 22 April, 2018). References to peer-reviewed journal articles on the E3SM atmosphere model are expected be added to the website.

**Reply to Reviewer #2**

**April 12, 2018**

We greatly appreciate the referee's careful and insightful review. Our responses are detailed below.

*I recommend publication after addressing concerns cited below. The paper is a useful description of water conservation errors in EAM V1.*

*Suggested word changes. if these suggestions are not correct then there are more serious problems that I do not understand in this paper.*

- *1-16: ".. errors in early V1 versions decrease .."*

- *1-17: "Increased vertical resolution in V1 results .."*

- *1-20: ".. beneficial for V1."*

- *2-18: ".. errors in V1$\alpha$ and .."*

- *3-3: " .. 30 everywhere .." I assume this model does not use step-mountain.*

- *3-15: ".. 6 sub-steps of 5 minutes each as .."*

- *5-1: "..chemical processes that operate on a single vertical column."*

- *5-7: "..passed on .."*

- *6-11: ".. which includes advection of air mass, momentum, and heat."*

- *8-20: ".. evaporation and ignoring the heat of sublimation."*

- *8-32: ".. clipped amount of downward moisture .."*

- *9-2: ".. received by the sub-surface components .."*

- *9-3: ".. are unclipped, .."*

- *9-9: "This can be seen .."*

The suggested wording changes are applied in the revised manuscript.

*Confusing things.*

*3-2: 90\*90\*6 = 48600. Why are there 2 additional columns on the cube sphere grid.*

The physical parameterizations are computed at the vertices (instead of quadrilateral faces) of the cubed sphere grid. 48600 ($90\times90\times6$) is the number of quadrilaterals, while $48600 + 2 = 48602$ is the number of vertices. The relationship between these two numbers can be derived using Euler's polyhedron formula. For example, the cube has 6 faces, and $6 + 2 = 8$ vertices. This is explained in a footnote on page 3 of the revised manuscript:

*"A spectral element contains $3 \times 3 = 9$ quadrilaterals, giving a total of $9 \times 30^2 \times 6 = 48600$ quadrilateral faces at ne30. The parameterizations are calculated at the vertices of the cubed sphere grid. Based on Euler's polyhedral formula, the number of vertices on a cubed sphere grid equals the number of quadrilaterals plus 2. Therefore, the total number of grid points for parameterization calculation is 48602".*

*3-12: How thick are the 14 layers over mountains?*

The original sentence in the discussion paper reads "...the vertical resolution is increased to 72 layers for both the dynamics and physics, with about 14 levels between the surface and 850 hPa." We replaced the second half of the sentence by:

*"with a typical layer thickness of 50–120 m in the bottom 1 km of the atmosphere, except for the lowest model layer which is about 20 m thick".*

These new numbers are valid both over the ocean and over the mountains.

*6-Fig2: It should be mentioned that this diagram shows processes of the stadium shaped cells of Figure 1, that (a) relates to se_type = 1, and (b) relates to se_type 0. From what I understand, that are 3 se_rsplit steps for each se_nsplit which makes the diagram misleading. I suggest you discard Figure 2 and explain its contents in the text.*

We revised Fig. 1 and Fig. 2 and the corresponding text to address this comment:

[Figure]

**Figure 1.** Diagrams showing the sequence of calculation (i.e., the time integration loop) in EAM. Left: V0. Right: V1γ. The blue stadium shapes refer to the resolved-scale dynamics and transport, and the diamonds refer to the exchange of mass and energy with other model components (e.g., land and ocean) through the coupler. The rectangular cells are parts of the physics package that describe the subgrid-scale physical and chemical processes. The colored boxes indicate parts of EAM that affect the concentrations of water species; these include the numerical fixers, deep and shallow convection, turbulent transport, and stratiform cloud macro- and microphysics. The coupling between resolved dynamics and parameterized physics is explained in Section 3.1 and illustrated in Figure 2. The flow chart of V1α is shown in Figure A1.

[Figure]

**Figure 2.** Diagrams showing two options for the coupling between parameterized physics and resolved dynamics in EAM. "Clipping" in panel (b) refers to the fact that negative water concentrations are reset to zero after the physics tendencies are applied. The examples shown here correspond to se_nsplit = 2, se_rsplit = 3. Further details are explained in Section 3.1.

*6-11 to 7-3: Clean this up. According to Table 4 there are 2 to 3 r steps for each n step. Make this clear. Mention that humidity advection is grouped with dynamical advection or with tracers depending on model version.*

The corresponding paragraph is rewritten to describe the two levels of sub-stepping:

*"The dynamical core calculates the advection of momentum, heat, air mass, and the mass of additional trace species such as water vapor, cloud condensate, aerosols and their precursors. Within the PDC interval of $\Delta t$, there are two levels of sub-stepping that are relevant to discussions in this paper: At the top level, the entire dynamical core is sub-cycled with se_nsplit (typically 2– 4) steps, each containing the calculating of horizontal advection followed by vertical remapping. The horizontal advection is further sub-cycled with se_rsplit (typically 2–3) steps per one vertical remapping. In the examples shown in Fig. 2, there are se_nsplit = 2 dynamics sub-steps (blue stadium shapes) each containing se_rsplit = 3 horizontal advection steps (green boxes)."*

We point out during the description of se_ftype =2 that water vapor is grouped with other tracers (e.g., cloud condensate and aerosols):

*"se_ftype=2: a hybrid method that uses option se_ftype=0 (Fig.2b) for the fluid dynamics variables (temperature, winds, and surface pressure), while option se_ftype = 1 (Fig. 2a) is used for water vapor, liquid- and ice-phase condensate, and all other advected tracers. "*

*7-22 to 7-24: Does this sentence apply to all se-ftypes? If so, move it.*

Yes, the sentence describes a feature of the transport scheme and is valid for all se_ftypes. We clarify in the revised manuscript that the condition of positive element-total concentration is fulfilled by se_ftype = 1 and for the first dynamics sub-step of se_ftype = 0, but can be violated for the later dynamics sub-steps when se_ftype = 0 is used, resulting in a situation that is not anticipated by the transport scheme:

*"This condition is fulfilled by se_ftype = 1 and during the first dynamics sub-step of se_ftype = 0. However, for the later dynamics sub-steps of se_ftype = 0, our analysis indicates that the inconsistency between the intermediate model state and the physics tendencies of time step n can lead to negative element-total concentrations that are not anticipated by the transport scheme."*

*8-27: Say more about the time stepping method. Is it explicit? If it is implicit, is it intelligent enough no recognize the availability of water vapor in the boundary layers and not just the lowest layer?*

We agree with the referee that how the downward moisture flux affects near-surface humidity depends on how the flux is applied in the subsequent calculations. In the revised manuscript, we point out that QNEG4 assumes (1) the downward flux affects only the bottom layer, (2) no moisture source is provided from the layers aloft, and (3) an Euler forward method is used for time integration. QNEG4 is admittedly a very simplistic and aggressive limiter. Although experience has shown that *some* amount of adjustment in the downward moisture flux is needed, the actual amount applied by QNEG4 is likely an overestimation since the turbulence parameterizations in both E3SM V0 and V1 uses implicit time stepping methods. We also added text in the same subsection to note that the new QQFLX fixer determines the amount of water vapor to borrow based on the same 3 assumptions used by QNEG4, hence QQFLX probably also does more work than necessary. Future work on the coupling between surface fluxes and the turbulence parameterization could help to address this issue. The corresponding text has been revised to:

*"The surface flux parameterizations can occasionally predict strong downward moisture fluxes. Since EAM uses relatively long time steps for the physics package (30 min at 1° resolution), the downward fluxes could lead to negative humidity in the near-surface layers of the atmosphere model, depending on how the surface fluxes are applied in the subsequent calculations, e.g., whether they are applied as an immediate moisture sink in the lowest model layer or as the lower boundary condition in the turbulence scheme, and whether a simple explicit time stepping or a sophisticated implicit and positive semi-definite method is used.*

*E3SM inherited from its predecessor a subroutine called QNEG4 to guard against negative humidity. The subroutine limits the downward surface moisture flux to an amount that would result in zero water vapor in the lowest layer, assuming (1) the downward flux affects only the bottom layer, (2) no moisture source is provided from layers above, and (3) an Euler forward method is used for time integration. QNEG4 is admittedly a very simplistic and aggressive limiter. Although experience has shown that some amount of adjustment is needed for the downward moisture flux, the actual amount applied by QNEG4 is likely an overestimation since the turbulence parameterizations in both E3SM V0 and V1 use implicit time stepping."*

*9-22: Do these errors occur consistently over mountain tops of at ocean cells adjacent to mountains, or are they more sporadic? Same question applies to other sub-sections of Section 3.*

A paragraph is added to Section 5 ("Simulations and results") to address these questions:

*"In terms of geographical distribution (not shown), the PDC errors in V1α systematically occur in cloudy regions with strong horizontal gradient in cloud condensate. The LHFLX errors occur typically in middle and high latitudes due to the lower surface temperature there and the more frequent occurrence of ice sublimation/deposition. The QNEG4 errors mostly occur as isolated and sporadic large values over land, while the QNEG3 and INTERR errors are typically very small and randomly distributed in cloudy regions over the globe."*

*10-27: It appears that your water mass change is distributed uniformly over the globe. This water mass mainly affects the ocean fraction of the Earth. Should your sea-level changes be computed as ΔH / (1 - OceanFraction). Please comment on this.*

The water mass change we report in the paper is indeed uniformly distributed over the globe. In reality, the mass change is likely to end up in the oceans and in reservoirs over land such as lakes, ice caps and glaciers, hence our formula (Eq. 6) is likely to underestimate the resulting change in sea level. To provide an accurate assessment of the impact of the conservation error, one should conduct a pair of coupled model simulations with and without the fixes discussed in our paper, and compare the simulated sea levels. Unfortunately we did not have sufficient resources to conduct such simulations to evaluate the impact of water conservation in isolation. (The coupled simulations presented in the paper were not specifically conducted for the water conservation investigation, and therefore contained various other code changes that had impact on the simulated sea level.) The "equivalent sea level rise" reported in our paper is essentially a measure of water conservation error, not the actual sea level drift in E3SM. This is clarified in the revised manuscript after Eq. (6) is presented:

*"To demonstrate the long-term impact of the water conservation error, we also report on the global average of Eq. (4) and convert it to an "equivalent sea level change" using*

$$\Delta H^n = \frac{1}{\rho_l} \frac{\sum_{i=1}^{I} (A_i \Delta W_i^n)}{\sum_{i=1}^{I} (A_i)}. \tag{6}$$

*where $\rho_l$ is the density of liquid water. $\Delta H$ is the depth of liquid water that would accumulate at the Earth's surface if the atmospheric water from spurious sources was converted to precipitation and distributed evenly over the surface of the globe. In the actual model, part of the spuriously created water might stay in the atmosphere, resulting in less amount reaching the surface than $\Delta H$, although our analysis suggested that the long-term global-mean net surface flux $(P - E)$ was close to the spurious water source $\Delta W/\Delta t$. Another point to consider is that in reality, the increased precipitation is likely to end up in the oceans and in reservoirs over land such as lakes, ice caps and glaciers, hence the division in Eq. (6) by the surface area of the entire globe is likely to underestimate the change in sea level. To provide an accurate assessment of the impact of the conservation error, one should conduct a pair of coupled model simulations with and without the fixes discussed in the paper, and compare the simulated sea levels. Unfortunately we did not have sufficient resources to conduct such simulations and evaluate the impact of water conservation in isolation. (The coupled simulations presented in this paper contained various other changes that have impact on the simulated sea level.) Therefore, the $\Delta H$ reported here should be interpreted as a measure of water conservation error rather than the actual sea level drift in E3SM."*

*12-Figure3: Most noticeable in the lower left panel, do these spikes originate from large errors at single grid cells? Although water mass borrowing has eliminated the spikes in model V1γ, the original errors are still there.*

Yes, the spikes seen in the QNEG4 error originate from large errors at single grid cells over land. This is mentioned in a new paragraph added to Section 5 (see also our response to the referee's comment on line 9-22 above).

The last sentence of the abstract and the last paragraph in the conclusions section of our paper both point out that the proposed fixers are remedies rather than root cure of the conservation problems. Future improvements in the time integration methods would be beneficial for the V1 model.

*17-14 to 17-16: "increased" or "decreased" relative to what. Perhaps this sentence should start: "Comparing V1a model at 2.8° versus that at 1°: the .."*

The comparison was made with respect to the 1° simulation. The corresponding sentence is revised in the new manuscript.

[revised manuscript text omitted]

~~Within each integration cycle of step size $\Delta t$, we denote the old and new time steps with indices $n$ and $n+1$, respectively. Several levels of nested sub-stepping are implemented for the dynamics. A "dynamics sub-step" (i.e., a dashed frame in Fig. 2) uses a step size of $\Delta t/se\_nsplit$. A dynamics sub-step contains $se\_rsplit$ sub-cycles of horizontal tracer advection followed by vertical remapping. Each horizontal advection sub-step (with a step size of $\Delta t/se\_nsplit/se\_rsplit$) is further divided into~~

[Figure]

**Figure 2.** Diagrams showing two options for the coupling between parameterized physics and resolved dynamics in EAM. "Clipping" in panel (b) refers to the fact that negative water concentrations are reset to zero after the physics tendencies are applied. The examples shown here correspond to se_nsplit = 2, se_rsplit = 3. Further details are explained in Section 3.1. ~~Note that in panel (b), for each dynamics sub-step (the dashed frame), a step size of $\Delta t / se\_nsplit$ (instead of $\Delta t$) is multiplied to the physics tendencies to update the atmospheric state. $se\_nsplit$ is the number of sub-steps for dynamics (including vertical remapping of the semi-Lagrangian vertical coordinate). $se\_rsplit$ is the number of sub-steps for tracer advection in each dynamics sub-cycle. Numbers in the parentheses indicate the model time step (n or n+1) and sub-steps (i–ix).~~

~~stages of the Runge-Kutta time integration and even smaller steps of numerical diffusion, but those do not affect water species and thus are not depicted in Fig. 2. Within a dynamics sub-cycle of step size $\Delta t/se\_nsplit$, the horizontal advection sub-steps and the vertical remapping are calculated sequentially, meaning that each calculation is based on the atmosphere state updated by the previous calculation. Those intermediate atmosphere states are labeled with Roman numbers in parentheses in Fig. 2.~~

[revised manuscript text omitted]

---

## Author Response (AR2)

Dear Editor,

We updated the URL in the "code availability" section and provided a link for each code version (tag) used in this study.

Sincerely,

Kai Zhang